# Ubiquitous increases in flood magnitude in the Columbia River Basin under climate change

Laura E. Queen[1], Philip W. Mote[1], David E. Rupp[1], Oriana Chegwidden[2], and Bart Nijssen[2]

[1]Oregon Climate Change Research Institute, Oregon State University, Corvallis OR 97331 USA

[2]Department of Civil and Environmental Engineering, University of Washington Seattle WA 98105 USA

*Correspondence to*: Laura Queen (lqueen@uoregon.edu)

**Abstract.** The US and Canada have entered negotiations to modernize the Columbia River Treaty, signed in 1961. Key priorities are balancing flood risk, hydropower production, and improving aquatic ecosystem function while incorporating projected effects of climate change. In support of the US effort, Chegwidden et al. (2017) developed a large-ensemble dataset of past and future daily streamflows at 396 sites throughout the Columbia River Basin (CRB) and select other watersheds in western Washington and Oregon, using state-of-the art climate and hydrologic models. In this study, we use that dataset to present new analyses of the effects of future climate change on flooding using water year maximum daily streamflows. For each simulation, flood statistics are estimated from Generalized Extreme Value distributions fit to simulated water year maximum daily streamflows for 50-year windows of the past (1950-1999) and future (2050-2099) periods. Our results contrast with previous findings: we find that the vast majority of locations in the CRB are estimated to experience an increase in future streamflow magnitudes. The near-ubiquity of increases is all the more remarkable in that our approach explores more possible differences than previous studies; however, like previous studies, our modeling system was not calibrated to minimize error in maximum daily streamflow, and may be affected by unquantifiable errors. We show that on the Columbia and Willamette rivers, increases in streamflow magnitudes are smallest downstream and grow larger moving upstream. For the Snake River, however, the pattern is reversed, with increases in streamflow magnitudes growing larger moving downstream to the confluence with the Salmon River tributary, and then abruptly dropping. We decompose the variation in results attributable to variability in climate and hydrologic factors across the ensemble, finding that climate contributes more variation in larger basins while hydrology contributes more in smaller basins. Equally important for practical applications like flood control rule curves, the seasonal timing

of flooding shifts dramatically on some rivers (e.g., on the Snake, 20th century floods occur exclusively in late
spring, but by the end of the 21st century some floods occur as early as December) and not at all on others (e.g.
the Willamette).

## 1 Introduction

Among natural disasters in the Northwest, flooding ranks second behind fire in federal disaster declarations[1] since
1953 despite extensive flood prevention infrastructure. The largest flood in modern times on the Columbia
occurred in late spring (May-June) 1948, and obliterated the town of Vanport which lay on an island between
Portland, OR and Vancouver, WA, permanently displacing its 18,500 residents[2]. Other disruptive floods in the
region include the Heppner flood in 1903, one of the deadliest flash floods in US history (Byrd, 2014); floods on
the Chehalis River in both December 2007[3] and January 2009[4] that closed Interstate 5, the main north-south
transportation corridor through the Northwest, for several days each time at a cost of several \$m per day to freight
movement alone; and floods on the Willamette River in February 1996 and April 2019. The timing of typical
floods varies widely across the region: low-elevation basins in western Washington and Oregon typically flood
in November through February, whereas the snow-dominant basins east of the Cascades more typically flood in
spring, sometimes as late as June (Berghuis et al. 2016).

The Columbia River drains much of the Northwest, with the fourth largest annual streamflow volume in the US
and a drainage that includes portions of seven states plus the Canadian province of British Columbia (BC), an
area of 668,000 km$^2$ (Fig. 1). Its numerous federal and nonfederal dams provide flood protection, hydropower
production, navigation, irrigation, and recreation services. A treaty between the US and Canada, signed in 1961,
codified joint management of the river's reservoirs (and funded construction of new reservoirs in BC) primarily

---

[1] https://www.fema.gov/data-visualization-summary-disaster-declarations-and-grants accessed 8/6/2019

[2] https://www.oregonlive.com/portland/2017/05/vanport_flood_may_30_1948_chan.html accessed 8/6/2019

[3] https://www.seattletimes.com/seattle-news/extensive-flooding-3-confirmed-deaths-hundreds-of-rescues/ accessed 8/6/2019

[4] https://www.seattletimes.com/seattle-news/despite-drying-cooling-trend-flooding-and-road-closures-continue/ accessed 8/6/2019

to provide flood protection and hydropower production[5]. The US and Canada have entered negotiations to update the treaty; the USA's "key objectives include continued, careful management of flood risk; ensuring a reliable and economical power supply; and improving the ecosystem in a modernized Treaty regime." (*ibid*.) Both countries have expressed an intention to include the effects of climate change on  streamflows, and clearly a key aspect of hydrologic change is to inform the treaty negotiations of the influence of climate change on the magnitude of flooding.

While rising temperatures potentially affect all parts of the hydrologic cycle, in a snowmelt-dominated hydrologic system such as many of the Northwest's river basins, warming directly affects snow accumulation and melt (e.g., Hamlet et al. 2005). Observational studies have shown consistent changes toward lower spring snowpack (Mote et al. 2018), earlier spring streamflow (Stewart et al. 2005), and lower summer streamflow (Fritze et al. 2011) since the mid-20th century. Observations of trends in flooding in the US have generally failed to find any consistent trends (Lins and Slack 1999; Douglass et al. 2000; Sharma et al. 2018). Sharma et al. (2018) offer several possible explanations, chiefly "decreases in antecedent soil moisture, decreasing storm extent, and decreases in snowmelt". The detection of trends in floods is complicated by the interaction of extreme events and nonstationarity (Serinaldi and Kilsby, 2015). Moreover, as a result of the substantial alteration of rivers to prevent flooding (e.g., by the construction of dams and levees) during the observational period, the best long-term records - i.e., on streams with the least modifications - are on rivers that were not producing sufficiently disruptive floods to lead decision-makers to construct flood protection structures. That is, as flooding of settlements, infrastructure, or other assets led to the investments in flood protection structures on most rivers, thereby altering the streamflow regime and dividing any gauged records into pre- and post- modification, the ones that were left unmodified tended to be small and/or remote.

To interpret the ambiguous results from observed trends, Hamlet and Lettenmaier (2007) used the Variable Infiltration Capacity (VIC) hydrologic model forced twice with detrended observed daily weather for the period 1916-2003, with about 1°C of temperature difference between the two. They then compared 20- and 100-year flood quantiles for basins at varying sizes in the western US and found a wide range of changes in flood magnitude ranging from large decreases to large increases (+/- 30%).  Broadly, the responses depended somewhat on basin winter temperature, with the coldest basins (<-6°C) showing reductions in flood magnitude owing to reduced

[5] https://www.state.gov/columbia-river-treaty/ accessed 8/6/2019

snowpack, basins with moderate temperatures exhibiting a wide range of changes, and rain-dominant (>5°C)
basins showing little change, though the warm basins in coastal areas of Washington, Oregon, and California
showed increased flood magnitude.

Modeling work using state-of-the-art hydrologic models has been applied to understand where and how flood
magnitudes may change in the future. Tohver et al (2014) found widespread increases in flood magnitudes,
especially in temperature-sensitive basins (mainly on the west side of the Cascades), but their approach used
monthly GCM output so changes in daily precipitation would not be represented. Salathé et al. (2014) used a
single global climate model (GCM), the ECHAM5, linked to a regional climate model to obtain high-resolution
(in space and time) driving data for VIC over the period 1970-2069. As did Hamlet and Lettenmaier (2007), they
compared the ratio of flood change (2050s vs 1980s) against mean historical winter temperature and found a
majority of locations with a higher 100-year flood, in some cases by a factor of 2 or more; while they projected
increases in every one of the warmer basins (>0°C), a substantial fraction of colder locations had decreases in
flood magnitude.

Chegwidden et al. (2019) describe the process used to generate the streamflow ensemble used here. In addition,
they used analysis of variance (ANOVA) to analyze the different influences of choices of emissions scenario (as
a Representative Concentration Pathway - RCP), GCM, internal (unforced) climate variability, downscaling
method, and hydrologic model, and how those influences varied spatially across the domain and also seasonally
and by hydrologic variable. They found that the RCP and GCM had the largest influence on the range of annual
streamflow volume and timing, and hydrologic model had the largest influence on low streamflows. The
hydrologic variables they considered were snowpack (maximum snow water equivalent and date of maximum
SWE), annual streamflow volume, centroid timing (the date at which half the water year's streamflow has passed),
and seasonal  streamflow volume; primary focus was on centroid timing, annual volume, and minimum 7-day
streamflow. They did not examine high-flow extremes that can lead to flooding. The purpose of this paper is to
address this important gap in our understanding of the future Northwest hydrology; to do so, we use the largest
available ensemble of climate-hydrology scenarios. By using a large ensemble, we ensure a reasonable breadth of
climatic and hydrological futures in order to better describe the range of possible future flooding and how it varies
across the region with its diverse hydroclimates. We also note possible shortcomings associated with modeling
future flooding.

## 2 Methods

### 2.1 Hydrologic modeling data set

To assess changing flood magnitudes under climate change, we analyzed changes in water year maximum daily streamflows in a large ensemble of streamflow simulations at 396 locations in the CRB (Figure 1) and select watersheds in western Oregon and Washington (Chegwidden et al., 2017). The simulations were constructed from permutations of modeling decisions on forcing datasets and hydrologic modeling. Specifically, choices included two RCPs (RCP4.5 and RCP8.5), ten GCMs, two methods of downscaling the climate model output to the resolution of the hydrologic models, and four hydrologic model implementations, for a total of 160 permutations. For our analysis, we extracted a more tractable dataset of 40 simulations per location, by only considering simulations with RCP 8.5 and the Multivariate Adaptive Constructed Analogs (MACA) downscaling method (Abatzoglou and Brown, 2012).

The rationale for using a subset of the available data is as follows. First, the time-dependent set of greenhouse gas concentrations in RCP4.5 is fully included in RCP8.5, so any concentration of greenhouse gases on the RCP4.5 path can be converted to a point on RCP8.5 (at a different time). We analyzed results for both RCP8.5 and RCP4.5, and found that to first order the changes in flood magnitude in RCP4.5 were approximately 2/3 those in RCP8.5, which is also roughly the ratio of global temperature change over the period considered (IPCC Summary for Policymakers, 2014). For clarity we show only the results for RCP8.5. Second, we considered only simulations using the MACA downscaling method because of the method's ability to capture the daily GCM-simulated meteorology critical for assessing changes in extremes and its skill in topographically complex regions (Lute et al., 2015). The other downscaling approach used by Chegwidden et al. (2019), the Bias Correction and Statistical Downscaling (BCSD) method (Wood et al. 2004), produces probability distributions of daily precipitation inconsistent with the GCM response to forcings because the method stochastically disaggregates monthly data to daily data based on historical statistical properties of the daily data. This statistical property limits the ability of BCSD to reproduce changes in storm frequency in the future, making it a less attractive choice for daily extreme streamflow analysis (Hamlet et al. 2010; Guttman et al. 2014).

Model output used in this study came from the following ten GCMs: CanESM2, CCSM4, CNRM-CM5, CSIRO-Mk3-6-0, GFDL-ESM2M, HadGEM2-CC, HadGEM2-ES, Inmcm4, IPSL-CM5A-MR, and MIROC5. These ten GCMs were chosen primarily for their ability to accurately reproduce observed climate metrics during the

historical period mainly of the Northwest US but also at sub-continental and larger scales as assessed in Rupp et
al. (2013) and RMJOC (2018). The four hydrologic model implementations originated from two distinct
hydrologic models: the Variable Infiltration Capacity (VIC; Liang et al., 1994) model and the Precipitation Runoff
Modeling System (PRMS; Leavesley et al., 1983). VIC and PRMS are process-based, energy balance models and
were both run on the same 1/16th degree grid with output saved at a daily time step for the period 1950 to 2099.
VIC is a macroscale semi-distributed hydrologic model that solves full water and energy balances, and in these
simulations it also included a glacier model (Hamman & Nijssen, 2015). Three unique implementations of VIC
were used with independently derived parameter sets (P1, P2, P3) marked by differences in calibrated parameters,
calibration methodology, and meteorological and streamflow reference sets. PRMS is a distributed, deterministic
hydrologic model which, in contrast to VIC, does not allow for subgrid heterogeneity. See Chegwidden et al
(2019) for details. It is important to note that these hydrologic simulations and calibrations do not include reservoir
models and have not been calibrated for daily, let alone maximum daily, flows, and these shortcomings may affect
the results.
**2.2 Flood magnitude**
We assessed changes in flood magnitude in the Columbia River Basin by comparing water year maximum daily
streamflows over a 150-year period (1950-2100). We estimated the 10, 5, 2, and 1% probability of occurrence
(commonly referred to as the 10-, 20-, 50-, and 100-year flood, respectively) by fitting generalized extreme value
(GEV) probability distributions to simulated water year maximum daily streamflows for 50-year windows of the
past (1950-1999) and future (2050-2099) periods; see Figure 2 for an example. (We also looked at 30- and 75-
year windows, choosing 50 years as a balance between sample size favoring longer periods, and nonstationarity
considerations favoring shorter periods.) We used Python's scipy.stats.genextreme module (Jones et al., 2001) to
fit a Gumbel distribution and estimate flood magnitudes for each return period. We assessed change in flood
magnitude as the "discharge ratio" of the estimated future to past floods for a given return period; a ratio greater
than 1 indicates an increase in flood magnitudes while a ratio less than 1 indicates a decrease.

We describe how changes in flood magnitude vary by climatic zone across the PNW by using an efficient and
internally consistent proxy for climatic zone: the centroid of timing – the day in the water year that half the annual
volume of water has passed the stream location. The centroid of timing is a metric of snow dominance (e.g.,
Stewart et al. 2005) which is related to the spatial distribution of temperature and tends to decrease downstream.
This temporal proxy of a hydrologic characteristic is effective in the Columbia Basin where most of the
precipitation occurs in winter and the relative magnitude and timing of the freshet from the spring thaw is a good
indicator of importance of snowmelt to streamflow. An early centroid indicates that rain, which falls
predominantly during the cooler, earlier part of the year, is the driver of the peak streamflows at the location,
while a late centroid indicates that snowmelt during later spring months is the prime hydrological driver. We
computed the centroid using the 1950-79 simulated years. Note that Chegwidden et al. (2019) also used the *change*
in centroid as a hydrologic variable of interest; below, we discuss our results in the context of their findings.

**2.3 Model evaluation**
Comparing directly between gauged flows and modeled flows is inadvisable since the observed streamflows are
substantially altered by regulation, which is not accounted for in the hydrological model. However, a set of
streamflows called No Reservoirs No Irrigation (NRNI; RMJOC 2017) has been developed by federal agencies
to support practical analysis. The NRNI dataset exists at ~190 sites across the Columbia River Basin for the years
1928-2008, and streamflows are adjusted to correct for reservoir management and the diversions and evaporation
associated with both the reservoirs and with irrigated agriculture. This dataset is suitable for comparisons with
our modeling setup, and we have computed return period curves using GEV fits at all the NRNI locations (not
shown) for the period common to both NRNI and our ensemble, viz., 1950-2008. From these fits we have
estimated the 10-year and 100-year values (Figure 3). On the lower mainstem Columbia (Figs 3a and d), the return
period curves are very close to those computed from NRNI and the means of simulations are almost all within 8%
of the NRNI values. Individual hydrologic model configurations are not consistently biased across the basin nor
across return periods; despite its different provenance, PRMS generally lies within the return period streamflows
of the three VIC configurations rather than being consistently different from all VIC configurations, although the
lowest values are from PRMS. On the Snake River, the mean of modeled high streamflows range from 5% above
NRNI at Little Goose to 24% above at Oxbow for 10-year floods (and 14% to 41% for 100-year) but again no
hydrologic model stands out as strongly biased. On the Willamette, however, the modeled 10-year and 100-year
flood magnitudes lie almost entirely below NRNI and the means are too low by from 30% (T. W. Sullivan, 10-
year) to 50% (Hills Creek, 100-year). PRMS and the P2 calibration of VIC are consistently closer to NRNI on the
Willamette. In general, the simulated flood statistics are least biased on larger river reaches where the hydrographs
are less flashy. For the Columbia mainstem, modeled extreme high streamflows agree well with the NRNI dataset.

We also examined the ensemble performance for 1950-2008 in the distribution of timing of peak daily streamflow
for 28 locations along the Columbia, Snake, and Willamette (a subset is shown in Figure 4). At all locations we
examined, the median date (as well as earliest and latest quartiles) of annual maximum daily streamflow in the
ensemble is within 10 days of the observed, from NRNI. The modeled distribution is shifted slightly later than
NRNI on the lower Columbia and slightly earlier than NRNI on the Willamette. As with magnitudes, the
agreement in timing suggests a robust modeling set-up since the comparison tests the ability of the combined
climate-hydrologic modeling system to match observed, constrained only by the broad physics of the climate
system and by meteorological bias correction (which cannot substantially change the timing of the day of the year
most conducive to high streamflows). Although the modeled streamflows are calibrated, the statistical approach
to calibrations is not sensitive to the extreme maximum daily streamflow studied here.
It is worth stressing that these results compare outputs of hydrologic models in which the inputs are simulated
daily weather (which is then bias-corrected) rather than observed daily weather, and that the hydrologic models
are calibrated to 7-day means rather than the daily values relevant here. In other words, we are evaluating the
ability of the *combination* of simulations of weather and hydrologic response. The weaknesses evident in Figure
4 pose a note of caution in interpreting our results, but a full diagnosis of the causes of the shortcomings (especially
on the Willamette) is beyond the scope of this paper, as is the evaluation of our modeling system's performance
at other locations besides these rivers.

**3 Results**

**3.1 Regional changes in flood ratio**

Figure 5 shows the changes in maximum daily discharge for all of the 396 streamflow locations for different
return periods. The horizontal position of each circle represents the centroid of timing. The circles are semi-opaque
so overlapping circles lead to a deeper saturation. Points on the same river are ordered from more to less snow
dominant (i.e., right to left) traveling downstream; strings of circles in a smooth pattern usually indicate one of
the larger rivers, highlighted in Figure 6.  Each circle in Figures 5 and 6 represents an average of 40 simulations:
10 GCMs and 4 hydrologic model configurations.
A striking result in Figure 5 is that, in contrast to the results of Tohver et al. (2014), the flood magnitude increases
(i.e., the discharge ratio exceeds one) at nearly every streamflow location and return period (though not for every
individual climate scenario, as shown in Figure 7). Broadly, the patterns are similar across all return periods
though with slightly higher ratios for longer return periods, and subsequent figures will show only the 10- and
100-year floods. For the streamflow locations with centroid <125 or so (i.e. February 2), flood ratios are fairly
concentrated about 1.25 for all return periods. For mixed rain-snow basins, roughly delineated by centroids
between 125 and 160 (March 8 most years), flood ratios range widely from just below 1 to about 2.4 for the 10-
year and 3.2 for 50- and 100-year floods. For the longer return intervals, there is a wide range of projected changes
in daily flood at many locations (indicated by the red coloring). This is undoubtedly partly due to the GEV fit
extrapolating from 50 to 100 years. Finally, for the basins with streamflow centroid >160, the ratios have a smaller
range, from slightly greater than 1 to a maximum that increases from about 2 for the 10-year, to about 2.75 for
100-year. Tohver et al. (2014) distinguished basins by their DJF temperature, a rough proxy for our snow
dominance metric, and found a substantial number of locations where the flood ratio for both 20-year and 100-
year flood was as much as 20% lower for the 2040s compared with a historical period. We return to this point in
the                                                                                                    conclusions.
To understand better how flood magnitude changes along the length of a river, we focus (Figure 6) on a handful
of significant rivers in the region: the mainstem Columbia, Willamette (along with major tributaries the McKenzie
and Middle Fork Willamette), and Snake, and also on the Chehalis in southwest Washington (see Introduction).
Flow locations and select numerical results are listed in Table 1. Many of the larger tributaries also have
streamflow points in our dataset, so we can infer the role of tributaries in changing the flood magnitudes in the
future, as discussed below. The Columbia River includes the most snow-dominant basins, with a centroid of >190
days (early to mid April) in the Canadian portion of the basin. The flood ratio decreases almost uniformly along
the length of the river, from 1.3 for the 10-year and >1.5 for the 100-year in the Canadian portion to just above 1
at the last few points along the river (The Dalles, Bonneville, and Portland). Past flood events on the mainstem
Columbia are exclusively associated with large spring snowmelt, and the large tributaries (the Yakima, Snake,
and Willamette) contribute annual streamflow volume but rarely contribute peak streamflow at the same time; as
shown below, the future flood timing changes but flood magnitudes change little in the lower Columbia owing to
the fact that the Columbia integrates such diverse hydroclimates.  Like the Columbia, the Willamette also has
flood ratios that decrease along the length of the river as it integrates more diverse hydroclimates, from 1.7 to 1.35
for both return periods. The McKenzie River (points 15-17), one of the three tributaries that converge at Eugene
to form the Willamette, is a highly spring-fed river with higher baseflow than is represented in the hydrologic
models, though it is unclear how that difference would manifest in the flood statistics. Nonetheless, the
combination of an important unrepresented process and the large errors in flood magnitudes relative to NRNI
(Figure 3) are potentially problematic for simulating future changes in flooding.

In contrast to the Columbia and the Willamette, the Snake behaves oppositely: flood ratio increases downstream
along the length of the river, until the confluence with the Salmon River, which drains a large mountainous area
of central Idaho. On parts of the Snake the ratios are as high as 1.4 for 10-year and 1.6 for 100-year. Then after
the confluence with the Salmon River, which has much lower change in discharge ratio, the ratios on the Snake
drop to about 1.2 for 10-year and about 1.3 for the 100-year. Our hypothesis is that in the Snake above the Salmon
River, the tributaries shift from snow-dominant to rain-dominant, so that a single storm can drive large rainfall-
driven increases (possibly with a snowmelt component) leading to larger synchronous discharges. The Salmon
and Clearwater rivers retain less exposure to such shifts, and dilute the effects of single large storms on flooding.

Each circle in Figures 5 and 6 represents an average of 40 simulations: 10 GCMs and 4 hydrologic model
configurations. To better understand the range in results, Figure 7 shows the discharge ratio for all 40 simulations
at each point on the mainstem Columbia. Although the mean flood ratio at the lowest two points is only barely
above 1, several ensemble members have ratios less than one, and a few have ratios >1.5. Moving upstream, the
range in results increases, as shown also by the color of the dots.
**3.2 Dependence of results on modeling choices**
As in Chegwidden et al (2019), we separate the results - here for the three largest rivers - into variations across
GCM (Figure 8) and variations across hydrologic model configurations (Figure 9). The ranking of flood ratios by
GCM changes substantially between basins and even within a basin, and does not correspond to the changes in
seasonal precipitation. For the upper Columbia River, the models with the least warming - inmcm4 and GFDL-
ESM2M (Rupp et al 2017) - have almost no change in flood magnitude, but the HadGEM2-ES which warms
considerably in summer produces a large decrease in flood magnitude. In the Willamette and Snake Rivers, the
range of projected flood changes by different GCMs remains large from the headwaters to the mouth of the river,
whereas for the Columbia the range diminishes considerably as one moves downriver.

The variation of results depends less on hydrologic model than on GCM (Figure 9), though the differences across
hydrological models are still substantial. For the Willamette, lower Snake, and both upper and lower Columbia,
the PRMS model predicts substantially larger increases in flooding than the three calibrations of the VIC model.
For the upper Snake, it predicts substantially smaller change than any VIC calibration. While it is perhaps not
surprising that the three calibrations of VIC are close to each other, it is striking just how different are the
projections from PRMS at most locations on these three rivers. Chegwidden et al. (2019) found that the main
contributors to differences in hydrologic variables (except low streamflows) generally were the climate scenarios
(GCM and RCP), consistent with our findings here. (The order of models is similar in the equivalent figure for
the 100-year return period, but we elected to show the 10-year figure since the 100-year figure is more difficult
to decipher because the symbols overlap with those from other rivers.)
To parse the contributions of climate factors (represented by the GCMs) and hydrologic factors (represented by
the hydrologic models), we perform ANOVA on the 40 discharge ratios. The pie charts in Figure 10 show the
proportion of the total variance explained by climate factors and hydrologic factors at different locations. For the
Willamette River, the portion of uncertainty connected to the climate grows more important and the portion of
uncertainty connected to the hydrologic variability less important going from the confluence of the three major
tributaries at Eugene to the mouth. For the Snake and Columbia rivers, climate is responsible for virtually all of
the variance in projections in the upper reaches, but only about half at the lowest point, similar to the Willamette.
The Willamette basin is much smaller, and a large storm can affect the entire basin on the same day (Parker and
Abatzoglou, 2016), whereas storms typically take a couple of days to move across the Snake and Columbia (and
generally move upstream). With larger and more diverse contributing areas, differences in the rates with which
the hydrological models transfer precipitation to the point of interest become more important. Unlike Chegwidden
et al. (2019), we did not attempt to isolate the response to anthropogenic forcing from internal climate
variability. Though several techniques for separating these two factors have been used (e.g., Hawkins and Sutton,
2009; Rupp et al., 2017; Chegwidden et al., 2019), these techniques are either infeasible with our dataset or we
question their suitability for the application to changes in extreme river flows.

**3.3 Change in timing**
Although in a broad hydrologic sense a flood is a flood regardless of what time of year it occurs, there are
potentially significant ecological differences depending on time of year; for example, scouring the river bottom
causing significant loss of salmon eggs (Goode et al. 2013). Moreover, water management policies are strongly
linked to the calendar year (see Discussion). We computed the probability of flooding for (all 40) past and future
simulations at all the points on the three rivers (Figure 6) as a function of day of year (Figure 11). For the
Willamette, no significant change in timing occurs; however, for the upper Willamette, a single peak in likelihood
in February becomes more diffuse. For the Snake, all locations see a shift toward earlier floods, consistent with
the transition to less snow-dominant and more rain-dominant. Whereas floods were historically concentrated in
the period of mid-May to mid-July, the projected future flooding period spans December to June. For the
Columbia, the mode in the flood timing shifts earlier by half a month in the upper Columbia to about a month in
the lower Columbia. The distribution also broadens with an elongated tail towards winter such that there is low,
but non-negligible, probability of floods occurring as early as January. The magnitudes of the 10- and 100-year
flood events in the lower Columbia are not projected to increase substantially (Figures 6-9). However, the window
during which a major flood could occur expands, with the likelihood of major flooding in May or April (or even
as early as February) increasing.

## 4 Discussion and conclusions

Our study joins a small number of others in examining high-flow extremes using a large hydroclimate ensemble.
Gangrade et al. (2020) used a similar ensemble approach analyzing hydrological projections for the Alabama-
Coosa-Tallapoosa River Basin with 11 dynamically downscaled and bias corrected GCMs (10 of which our
studies share) and 3 hydrologic models (including VIC and PRMS). While they did not examine extreme daily
streamflows, they did calculate changes in the 95th percentile of daily streamflow (Q95). Perhaps because of the
hydroclimatic uniformity of that basin, they found very small differences in Q95 across hydrologic models, which
contrasts with our results showing changes in flood magnitudes varying by watershed and distance downstream.
Thober et al. (2018) conducted a similar study in some European river basins, but rather than using a climate
ensemble they simply imposed uniform warming scenarios on a hydrologic model (i.e. a more straightforward
temperature sensitivity analysis rather than an exploration of the range of future climate scenarios). Other, smaller
ensemble studies of floods in different basins include Huang et al. (2018), with 4 GCMs and 3 hydrology models,
and Vormoor et al (2015) with several parameterizations of one hydrology model.

Returning to the Northwest, our findings contrast with earlier work. Salathe et al. (2014) found decreases in flood
magnitude at a substantial number of sites, but our results show increases in flood magnitude at nearly every
return period and location, which includes about 100 locations not included in their study. They also noted that
directly downscaling the GCM outputs leads to a smaller range of results than when running the regional model
as an intermediate step, so we infer that if we had had access to RCM simulations driven by all 20 of our RCP-
GCM combinations, our range of results might have been larger. Another important difference may be in the
spatiotemporal coherence of extreme precipitation, which in the RCM would be generated directly by the
interaction of synoptic-scale storms, topography, and to a small extent by surface water and energy balance; and
in our study, by the interaction of the GCM-scale synoptic storms and constructed analogs derived from
observations. A large ensemble would reduce the magnitude of that effect. In our study, the MACA statistical
downscaling approach preserves much of the daily variability from the GCM, so the primary reason for the
difference between our results and theirs is probably the fact that we analyzed 40 scenarios. Some locations, for
example the points on the lower Columbia river, had a handful of ensemble members with decreasing flood
magnitude. But averaging the entire ensemble nearly always resulted in an increase in flood magnitude. It is
possible therefore that their study, repeated with a larger ensemble of hydrologic-climate model combinations,
might have found ubiquitous increases in flood magnitude as ours did.

Prior results (Hamlet and Lettenmaier 2007, Tohver et al. 2014, Salathe et al. 2014) suggested a decrease in flood
magnitude in snowmelt-dominated basins like the Columbia, since reduced snowpack reduces the store of water
available to be released quickly in a spring flood (like the May-June 1948 Vanport flood). In a subbasin of the
Willamette, Surfleet and Tullos (2013) projected decreases in flood magnitude for return periods > 10 years in the
Santiam River basin under a high-emissions scenario (SRES A1B, 2070-2099 vs. 1960-2010; 8 GCMs),
attributing the decreases to fewer large rain-on-snow events. Our results for the Santiam River show an *increase*
of 40% for both 10- and 100-year floods; this result includes rain-on-snow events, since they are represented in
VIC, which computes the accumulation of water in the snowpack and determines whether sufficient energy has
been provided to create a melt event. Our results point to ubiquitous increases in magnitude throughout the basin,
even on the lower mainstem Columbia. We also project some large increases in flood magnitude in the coldest
basins, including the headwaters of the Columbia, suggesting that the former results were missing some key
details. It seems likely that any reduction in flood magnitude originating from the warming-induced reduction in
spring snowpack is offset by other factors. While there is evidence that warmer future temperatures could
engender slower melt rates (Musselman et al. 2017), the effect on high streamflow events is less clear. For
example, Chegwidden et al (2020) showed that magnitudes of both rain- and snowmelt-driven floods are likely
to increase across headwater basins in the Pacific Northwest through the 21st century. These results emphasize
the necessity of revisiting reservoir rule curves, which are strongly tied to historical hydrographs, and also
emphasize that changes in the seasonality of flooding can be dramatically different from the changes in the mean
hydrograph. In particular, in the lower Snake and lower Columbia, changes in magnitude of flooding are modest
but changes in timing of the earliest quartile of flood events is much larger than the 0.5-1 month shift in the mean
hydrograph.

The evaluation of the modeling system in section 2.3 raises some concerns about the reliability of our results,
especially as to flood magnitude on the Willamette mainstem, and also in smaller basins where we have not
performed an evaluation. While this is a concern in an absolute sense, in a relative sense our results are probably
more robust than those of earlier studies in the Northwest, for several reasons. First, previous studies have rarely
provided the sort of evaluation of flood statistics that we show in section 2.3. Second, we used more
methodological variations, which tend to broaden, not narrow, the spread of results, and yet we still obtained a
narrowing of the spread of results to almost ubiquitous increases. Third, our use of a large ensemble, which
samples a wide climate space by using GCMs as opposed to RCMs. Conventional wisdom and evidence from the
weather and seasonal climate forecasting realms illustrate the utility of considering ensembles, and that generally
the true outcome of a prediction lies near the middle of the ensemble. Our ANOVA analysis (Figure 10) shows
that climate scenarios contribute a majority of the variation among results for most of the basin. Consequently, it
is of great importance to sample the climate scenarios broadly, which currently only GCMs can do. Large
ensembles of RCMs are rare; the 12-member NARCCAP ensemble (6 RCMs, 4 GCMs; Mearns et al. 2013), some
of whose model runs were completed a decade ago, remains the largest, but has a spatial resolution of only 50km.
CORDEX North America, similarly now has a comparable-size ensemble, but mostly still at 50 km (some at
0.22°), and was not available in such large numbers when we began our hydrologic simulations. At such spatial
resolutions, RCMs would still have to be further downscaled and bias corrected to use in our hydrologic models
(~6km spatial resolution). In the tradeoff between breadth of climate scenarios and spatial resolution, these
ensembles offer insufficient improvement in spatial resolution relative to our GCM ensemble to justify sacrificing
the breadth in climate scenarios represented by choosing just 4 GCMs. While RCMs certainly have their place in
such work and were used in some previous studies, using GCMs in this study allowed for a larger climate space
to be sampled, thus adding to the robustness of our results.


Although the likeliest outcome, as shown in Figure 7, is for smaller changes in flood magnitude in the lower
Columbia than elsewhere, a prudent risk management strategy would consider the range of possibilities. The
validation (Figures 3 and 4) provides no *a priori* basis for excluding or under-weighting the projections from any
hydrologic model. On the Willamette, a rain-dominant basin, our hydrologic simulations of flood magnitudes are
biased low; possible causes for the low bias originate both in the climate and hydrological models. For example,
a low bias in extreme daily precipitation may lead to an underestimation of the hydrologic response. We also note
that the hydrologic models were calibrated to 7-day means rather than daily values and may underestimate the
daily response in smaller basins. Nevertheless, three physical processes contribute directly to the increase in
magnitude: an increase in seasonal precipitation affecting soil saturation, an increase in extreme daily
precipitation, and a warming-induced reduction in the snow-covered area in the wet season. In our results for the
Willamette this reduction in snow-covered area reduces the buffering effect of snow accumulation during storms
and more than offsets an increase in melt from rain-on-snow events. This mechanism is supported by Chegwidden
et al (2020) who, using the same underlying dataset as our study, project a growth in both prevalence and
magnitude of rain-driven floods at the expense of floods from snowmelt and rain-on-snow events.

Our findings provide an initial indication of how existing flood risk management could respond to a warming
climate. Reservoir management is guided by rule curves which are intended to reflect the changing priorities and
risks during the year. For example, reservoirs used for flood control have rule curves that require reservoir levels
to be lowered when approaching the time of year when flood likelihood increases, and reservoir levels may be
raised as the likelihood decreases. For the Willamette, we found little change in the distribution of timing of flood
events, which indicate that with the state of the science today, reservoir rule curves may need to be altered as to
magnitude of flooding (which our results indicate will increase by 30-40%) but not timing; a reservoir model,
along with further investigation of the low bias in observed flood magnitudes (Figure 3e and 3f) would be required
for complete understanding of how flood risk (magnitude and timing) will actually change. For the Snake, larger
shifts in the timing imply a need to completely re-evaluate the existing rule curves. For the Columbia, the mode
in flood timing shifts earlier by half a month in the upper Columbia to about a month in the lower Columbia. The
distribution also broadens, with an elongated tail towards winter such that there is low, but non-negligible,
probability of floods occurring as early as January. These changes in timing imply a need for moderate alteration
of rule curves for reservoirs in the Canadian portion of the Columbia Basin.

Our results should not be taken as a precise prediction of flood magnitude change but rather as the best available
projections given the current state of the science. Two important factors need to be considered when interpreting
our results: first, in using RCP8.5, we selected the most extreme scenario of rising anthropogenic greenhouse gas
concentrations. If efforts to stabilize the climate before 2050 are successful, the flood magnitudes shown here will
undoubtedly be smaller (our analysis suggests most of the locations would see a change in flood magnitude about
1/3 smaller, for RCP4.5; e.g., a ratio of 1.3 (30% increase) for RCP8.5 would correspond to a ratio of 1.2 for
RCP4.5).

The second important factor in interpreting our results is that the actual river system in the Northwest includes many dams, a majority of which have flood control as a primary (or at least a top) objective. As a result, actual streamflows (and the changes in streamflow) at a given point in the river would be altered by reservoir management. Translating these changes in flood magnitude into actual changes would require a reservoir model for the basin or subbasin of relevance. One could then compute optimal rule curves for the major flood control reservoirs (perhaps time-evolving every couple of decades, to reflect the likely changes in scientific understanding and emissions trajectory). Even without that additional analysis, however, our results stress that the magnitude and/or timing of flood events will change throughout the basin. In other words, what worked for flood control in the past will not work as well in the future.

This study may have some utility in framing and quantifying the possible changes in flood risk as the Columbia River Treaty is in renegotiation, but further work would be needed to assign probabilities to future flood magnitude. Such work includes (a) a deeper understanding of the underlying model differences to explain differences in model sensitivities (our analysis in section 2.3 shows that PRMS performs about as well as the three calibrations of VIC for simulating past peak streamflows, but more work would be needed to understand the reasons for divergence in future projections), (b) applying different statistical and/or dynamical downscaling methods, and (c) using a more sophisticated approach to evaluating extremes in a nonstationary climate (as advocated by Serinaldi and Kilsby, 2015). The mechanisms of flooding in the upper Columbia and elsewhere are also a key question arising from this work; this and other work is needed to decipher the cause of the discharge ratio patterns we found along the major rivers. Furthermore, a new generation of GCM outputs (CMIP6, Eyring et al. 2016) already has data available from over 25 GCMs; in the near future, it would be feasible to apply a newer multi-model hydrologic modeling approaches (e.g., Clark et al., 2015) to the new generation of GCMs, though perhaps no significant changes would result.

Nonetheless, with current knowledge the fact that very few locations would see a decrease in flood risk under any climate/hydrologic scenario is a strong statement of the need to update all aspects of flood preparation: the definition of N-year (especially 100-year) return period streamflows, flood plain mapping, and reservoir rule curves, to name a few. Moreover, the challenges that the renegotiated Columbia River Treaty faces in accounting for climate change now appear to include the necessity of incorporating the likely increase in flood risk throughout the region.


Generally, this study shows how complex the spatial and temporal patterns of change can be in a mixed rain-and-
snow basin. Basins of similar size and hydrological response to warming exist on most continents, so our results
provide a warning against using a small number of climate scenarios or a single hydrologic model to estimate
changes in flood risk in other basins.


**Code/data availability.** The data used here are available at https://zenodo.org/record/854763.

**Author contribution.** L. Queen performed all analyses, wrote portions of the text, and edited the document. P.
Mote guided the analysis and wrote and revised much of the text. D. Rupp guided the analysis and edited the
document. O. Chegwidden generated the underlying dataset, guided the analysis, provided assistance with
programming, and commented on the text. B. Nijssen generated the underlying dataset and commented on the
text.

**Competing interests**. The authors declare no competing interests.

**Acknowledgments.** This project originated as a senior honors thesis by the first author, who thanks
Hank Childs of the University of Oregon for his mentorship. The research was supported by the
NOAA Climate Impacts Research Consortium, under award #NA15OAR4310145. We acknowledge
the World Climate Research Programme's Working Group on Coupled Modelling, which is
responsible for CMIP, and we thank each respective climate modeling group for producing and
making available their model output. For CMIP the U.S. Department of Energy's Program for Climate
Model Diagnosis and Intercomparison provides coordinating support and led development of software
infrastructure in partnership with the Global Organization for Earth System Science Portals.

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

**Figure captions.**

**Figure 1.** Domain of hydrologic simulations used in this paper, with colors indicating elevation of each grid cell, major rivers highlighted in blue, and numbers indicating locations of streamflow points highlighted in Figures 6-11, and Table 1. See Chegwidden et al. (2017, 2019) for all streamflow locations plotted in Figure 5.

**Figure 2.** Generalized Extreme Value fit of annual maximum daily streamflow from 50 years of simulation using output from one GCM (HadGEM2-ES), one hydrologic model (PRMS), for the Willamette River at Portland. Red and blue dots/ lines indicate the annual values and GEV fit for the 1950-99 'past' and 2050-99 'future' periods.

**Figure 3.** Comparison of 10-year (a, b, c) and 100-year (d, e, f) flood magnitudes from the observationally derived NRNI and the 40 climate-hydrologic model simulations, for 1950-2008, for select locations on the rivers as shown.

**Figure 4.** Statistical representations of the variation through the water year of the timing of flood events, 1950-2008, for NRNI (blue) and the 40 simulations of 1950-2008 with the climate-hydrology modeling system (green). To create each curve, the dates of the 5 highest streamflows in the period of record are tallied, and the resulting distributions smoothed. Long dashed lines indicate median date, short dashed lines the lowest and highest quartiles. MCD= Mica Dam (upper Columbia), TDA= The Dalles (lower Columbia, between the confluences of the Snake and Willamette), LGS = Little Goose (lower Snake), BRN=Brownlee, SVN=T. W. Sullivan (lower Willamette near Portland), DEX=Dexter (middle fork Willamette).

**Figure 5**. Discharge ratios (future:past) versus centroid of timing (day on which 50% of water-year streamflow has passed, an indicator of snow dominance) for all 396 locations and four return periods. For each location, the average of 40 ensemble member ratios calculated from GEV distribution fitting from 50-year windows for the future (2050-2099) and past (1950-1999) time periods is shown. Points are sized by average daily streamflow and colored by the coefficient of variation of the 40 ratios.

**Figure 6.** As in Figure 5 but only for points on the indicated rivers. Dashed lines indicate tributaries: 9-12 are on the Middle Fork Willamette, 15-17 on the McKenzie; tributaries of the Snake are the Grand Ronde (14), Clearwater (17) and Salmon (24). In the lower panel, the Grand Ronde and Salmon are clearly distinguished by a black circle around their perimeter. Table 1 translates the codes in the legend into named locations and shows the

numerical values represented in the figure. As is evident from both snow-dominance and size, locations are ordered downstream to upstream from left to right for each river.

**Figure 7.** Averaged (large circles) and individual ensemble member (small colored circles) discharge ratios for simulated streamflow locations along the mainstem Columbia River for the 10-year (top) and 100-year (bottom) return periods. As shown in the legend, the color of the dots distinguishes results by hydrologic model setup.

**Figure 8.** Average ratios of all 40 ensemble members (large circles) and the average of 4 hydrologic model results for each GCM (symbols), shown for simulated streamflow locations along the Willamette (top), Snake (middle), and the mainstem Columbia (bottom) for 100-year return periods. GCMs are ordered in the legend by their ranking in Rupp et al. (2017), representing their ability to simulate Northwest climate.

**Figure 9.** As in Figure 8 but averaged by hydrologic model, for 10-year return period, and combined into one panel.

**Figure 10**. ANOVA results for select locations on the indicated rivers, for climate and hydrologic factors (and the residual). Charts are numbered to correspond with their location in Figure 6, with the most-downstream location at the top.

**Figure 11.** Statistical representations of the variation through the water year of the timing of flood events. For each of the 40 simulations, the dates of the 5 highest streamflows in the 50-year past (blue) and future (green) windows are tallied, and the resulting distributions smoothed. Long dashed lines indicate median date, short dashed lines the lowest and highest quartiles.



**Table 1 Information about locations featured in this paper - location, river, and discharge ratios**

| River | UW Key | Description | 10-year flood discharge ratios | | | | 100-year flood discharge ratios | | | |
|---|---|---|---|---|---|---|---|---|---|---|
| | | | Avg. | Coeff. of Var. | Min | Max | Avg. | Coeff. of Var. | Min | Max |
| Chehalis | CHEGR | Chehalis R nr Grand Mound | 1.21 | 0.09 | 1.03 | 1.42 | 1.34 | 0.18 | 0.87 | 2.07 |
| Chehalis | CHE | Chehalis R at Porter | 1.21 | 0.08 | 1.03 | 1.40 | 1.31 | 0.16 | 0.91 | 1.89 |
| Willamette | SVN | T.W. Sullivan | 1.33 | 0.09 | 1.07 | 1.64 | 1.39 | 0.22 | 0.87 | 2.39 |
| Willamette | WILPO | Portland | 1.34 | 0.09 | 1.08 | 1.69 | 1.40 | 0.23 | 0.86 | 2.47 |
| Willamette | WILLA | Newberg | 1.34 | 0.09 | 1.09 | 1.66 | 1.40 | 0.22 | 0.88 | 2.44 |
| Willamette | SLM | Salem | 1.37 | 0.09 | 1.10 | 1.70 | 1.43 | 0.22 | 0.84 | 2.52 |
| Willamette | ALBO | Albany | 1.40 | 0.09 | 1.11 | 1.73 | 1.47 | 0.20 | 0.89 | 2.40 |
| Willamette | HARO | Harrisburg | 1.45 | 0.10 | 1.18 | 1.86 | 1.50 | 0.22 | 0.88 | 2.37 |
| Willamette | JASO | Middle fork @ Jasper | 1.50 | 0.14 | 1.20 | 2.13 | 1.57 | 0.23 | 0.93 | 2.68 |
| Willamette | DEX | Dexter | 1.55 | 0.16 | 1.17 | 2.33 | 1.61 | 0.22 | 1.05 | 2.67 |

| River | UW Key | Description | 10-year flood discharge ratios | | | | 100-year flood discharge ratios | | | |
|---|---|---|---|---|---|---|---|---|---|---|
| | | | Avg. | Coeff. of Var. | Min | Max | Avg. | Coeff. of Var. | Min | Max |
| Willamette | HCR | Hills Creek | 1.57 | 0.18 | 1.15 | 2.46 | 1.60 | 0.25 | 1.10 | 3.18 |
| Willamette | WILNF | Oakridge | 1.57 | 0.18 | 1.16 | 2.45 | 1.63 | 0.24 | 1.09 | 2.88 |
| Willamette | EUGO | WR at Eugene (NWP) | 1.50 | 0.12 | 1.26 | 2.04 | 1.54 | 0.22 | 0.88 | 2.57 |
| Willamette | WAV | Walterville | 1.54 | 0.13 | 1.29 | 2.13 | 1.55 | 0.18 | 1.04 | 2.23 |
| Willamette | LEA | Leaburg | 1.56 | 0.14 | 1.28 | 2.23 | 1.56 | 0.18 | 1.05 | 2.34 |
| Willamette | VIDO | McKenzie nr Vida | 1.57 | 0.15 | 1.28 | 2.32 | 1.58 | 0.19 | 1.02 | 2.41 |
| Willamette | COT | Cottage Grove | 1.25 | 0.11 | 0.97 | 1.69 | 1.39 | 0.29 | 0.78 | 2.38 |
| Snake | IHR | Ice Harbor | 1.20 | 0.13 | 0.92 | 1.75 | 1.26 | 0.28 | 0.79 | 2.84 |
| Snake | LMN | Lower Monumental | 1.20 | 0.13 | 0.92 | 1.76 | 1.26 | 0.28 | 0.78 | 2.77 |
| Snake | LGS | Little Goose | 1.19 | 0.13 | 0.92 | 1.77 | 1.26 | 0.28 | 0.78 | 2.83 |
| Snake | LWG | Lower Granite | 1.19 | 0.13 | 0.92 | 1.77 | 1.25 | 0.29 | 0.78 | 2.89 |

| River | UW Key | Description | 10-year flood discharge ratios | | | | 100-year flood discharge ratios | | | |
|---|---|---|---|---|---|---|---|---|---|---|
| | | | Avg. | Coeff. of Var. | Min | Max | Avg. | Coeff. of Var. | Min | Max |
| Snake | ANA | Anatone | 1.24 | 0.14 | 0.95 | 1.74 | 1.29 | 0.29 | 0.78 | 2.84 |
| Snake | LIM | Lime Point | 1.23 | 0.14 | 0.94 | 1.73 | 1.28 | 0.30 | 0.76 | 2.81 |
| Snake | HCD | Hells Canyon | 1.40 | 0.18 | 1.01 | 2.11 | 1.55 | 0.38 | 0.87 | 3.62 |
| Snake | OXB | Oxbow | 1.41 | 0.18 | 1.01 | 2.11 | 1.56 | 0.38 | 0.86 | 3.65 |
| Snake | BRN | Brownlee Dam | 1.41 | 0.18 | 1.01 | 2.12 | 1.56 | 0.37 | 0.86 | 3.63 |
| Snake | WEII | Weiser,ID | 1.39 | 0.18 | 1.02 | 2.09 | 1.53 | 0.35 | 0.86 | 3.28 |
| Snake | SNYI | Nyssa, OR | 1.40 | 0.18 | 1.04 | 2.16 | 1.52 | 0.33 | 0.89 | 3.21 |
| Snake | SWAI | Murphy, ID | 1.37 | 0.19 | 0.98 | 2.09 | 1.48 | 0.33 | 0.84 | 3.24 |
| Snake | CJSTR | CJ Strike Dam | 1.37 | 0.19 | 0.97 | 2.08 | 1.48 | 0.32 | 0.86 | 3.08 |
| Snake | SKHI | King Hill, ID | 1.37 | 0.19 | 0.96 | 2.08 | 1.48 | 0.32 | 0.85 | 2.84 |
| Snake | SNKBL WLSAL MON | Hagerman, ID | 1.35 | 0.18 | 0.93 | 2.05 | 1.46 | 0.31 | 0.83 | 2.66 |
| Snake | BUHL | Buhl, ID | 1.35 | 0.19 | 0.91 | 2.05 | 1.46 | 0.32 | 0.73 | 2.54 |

| River | UW Key | Description | 10-year flood discharge ratios | | | | 100-year flood discharge ratios | | | |
|---|---|---|---|---|---|---|---|---|---|---|
| | | | Avg. | Coeff. of Var. | Min | Max | Avg. | Coeff. of Var. | Min | Max |
| Snake | KIMI | Kimberly, ID | 1.33 | 0.19 | 0.89 | 2.03 | 1.44 | 0.33 | 0.74 | 2.47 |
| Snake | MILI | Milner, ID | 1.33 | 0.19 | 0.88 | 2.04 | 1.44 | 0.34 | 0.73 | 2.52 |
| Snake | MINI | Minidoka, ID | 1.33 | 0.19 | 0.86 | 2.02 | 1.45 | 0.33 | 0.70 | 2.53 |
| Snake | AMFI | Neeley American Falls | 1.32 | 0.19 | 0.85 | 1.99 | 1.45 | 0.34 | 0.67 | 2.69 |
| Snake | BFTI | nr Blackfoot, ID | 1.31 | 0.19 | 0.84 | 1.96 | 1.43 | 0.34 | 0.67 | 2.72 |
| Snake | SNAI | nr Blackfoot, ID | 1.30 | 0.19 | 0.84 | 1.95 | 1.43 | 0.34 | 0.67 | 2.69 |
| Snake | SHYI | Shelley, ID | 1.29 | 0.18 | 0.84 | 1.92 | 1.40 | 0.33 | 0.69 | 2.62 |
| Snake | LORI | Lorenzo, ID | 1.28 | 0.19 | 0.86 | 1.91 | 1.38 | 0.34 | 0.69 | 2.52 |
| Snake | HEII | Heise, ID | 1.28 | 0.18 | 0.86 | 1.91 | 1.37 | 0.33 | 0.70 | 2.53 |
| Snake | PALI | Irwin Palisades | 1.28 | 0.19 | 0.87 | 1.95 | 1.37 | 0.34 | 0.71 | 2.60 |
| Snake | JKSY | Jackson, WY | 1.26 | 0.15 | 0.89 | 1.73 | 1.35 | 0.30 | 0.80 | 2.46 |

| River | UW Key | Description | 10-year flood discharge ratios | | | | 100-year flood discharge ratios | | | |
|---|---|---|---|---|---|---|---|---|---|---|
| | | | Avg. | Coeff. of Var. | Min | Max | Avg. | Coeff. of Var. | Min | Max |
| Snake | SRMO | Moose, WY | 1.25 | 0.13 | 0.91 | 1.59 | 1.35 | 0.25 | 0.83 | 2.34 |
| Grand Ronde | TRY | Troy | 1.48 | 0.19 | 1.09 | 2.55 | 1.68 | 0.34 | 1.01 | 4.38 |
| Salmon | WHB | White Bird | 1.07 | 0.13 | 0.83 | 1.57 | 1.09 | 0.33 | 0.72 | 2.81 |
| Columbia | CRVAN | Vancouver | 1.03 | 0.09 | 0.90 | 1.22 | 1.05 | 0.13 | 0.80 | 1.49 |
| Columbia | BON | Bonneville | 1.03 | 0.09 | 0.90 | 1.21 | 1.05 | 0.13 | 0.80 | 1.49 |
| Columbia | TDA | The Dalles | 1.03 | 0.08 | 0.90 | 1.20 | 1.05 | 0.13 | 0.81 | 1.52 |
| Columbia | JDA | John Day | 1.02 | 0.08 | 0.90 | 1.19 | 1.05 | 0.13 | 0.80 | 1.51 |
| Columbia | MCN | McNary Dam | 1.02 | 0.08 | 0.89 | 1.18 | 1.05 | 0.13 | 0.80 | 1.45 |
| Columbia | CLKEN | Clover Island @ Kennewick | 1.03 | 0.10 | 0.82 | 1.22 | 1.11 | 0.14 | 0.84 | 1.49 |
| Columbia | CHJ | Chief Joseph | 1.06 | 0.11 | 0.83 | 1.25 | 1.15 | 0.15 | 0.85 | 1.70 |
| Columbia | GCL | Grand Coulee | 1.06 | 0.11 | 0.83 | 1.25 | 1.14 | 0.14 | 0.84 | 1.66 |
| Columbia | PRD | Priest Rapids | 1.04 | 0.10 | 0.82 | 1.22 | 1.11 | 0.13 | 0.84 | 1.54 |

| River | UW Key | Description | 10-year flood discharge ratios | | | | 100-year flood discharge ratios | | | |
|---|---|---|---|---|---|---|---|---|---|---|
| | | | Avg. | Coeff. of Var. | Min | Max | Avg. | Coeff. of Var. | Min | Max |
| Columbia | WAN | Wanapum | 1.04 | 0.10 | 0.82 | 1.22 | 1.11 | 0.14 | 0.84 | 1.58 |
| Columbia | RIS | Rock Island | 1.04 | 0.10 | 0.82 | 1.23 | 1.12 | 0.14 | 0.84 | 1.60 |
| Columbia | RRH | Rocky Reach | 1.05 | 0.10 | 0.83 | 1.23 | 1.13 | 0.14 | 0.84 | 1.61 |
| Columbia | WEL | Wells Dam | 1.05 | 0.10 | 0.83 | 1.24 | 1.14 | 0.14 | 0.85 | 1.63 |
| Columbia | ARD | Hugh Keenleyside (Arrow) | 1.13 | 0.12 | 0.87 | 1.43 | 1.24 | 0.21 | 0.69 | 1.83 |
| Columbia | RVC | Revelstoke | 1.19 | 0.12 | 0.91 | 1.62 | 1.36 | 0.23 | 0.69 | 2.08 |
| Columbia | MCD | Mica Dam | 1.22 | 0.12 | 0.94 | 1.66 | 1.41 | 0.24 | 0.72 | 2.12 |
| Columbia | DONAL | Donald | 1.28 | 0.14 | 1.02 | 1.79 | 1.55 | 0.25 | 0.94 | 2.38 |
| Columbia | CRNIC | Nicholson | 1.25 | 0.13 | 0.98 | 1.61 | 1.47 | 0.23 | 0.94 | 2.39 |
| Clearwater | SPD | Spalding, ID | 1.15 | 0.15 | 0.85 | 1.78 | 1.32 | 0.30 | 0.80 | 2.63 |
| Clearwater | DWR | Dworshak Dam, ID | 1.14 | 0.12 | 0.86 | 1.55 | 1.30 | 0.24 | 0.89 | 2.22 |
| Santiam | JFFO | Santiam R nr Jefferson | 1.40 | 0.10 | 1.14 | 1.81 | 1.41 | 0.25 | 0.81 | 2.27 |

| River | UW Key | Description | 10-year flood discharge ratios | | | | 100-year flood discharge ratios | | | |
|---|---|---|---|---|---|---|---|---|---|---|
| | | | Avg. | Coeff. of Var. | Min | Max | Avg. | Coeff. of Var. | Min | Max |
| Kootenay | COR | Corra Linn Dam, BC | 1.08 | 0.12 | 0.85 | 1.31 | 1.15 | 0.16 | 0.79 | 1.67 |
| Kootenai | LIB | Libby Dam, MT | 1.17 | 0.14 | 0.92 | 1.52 | 1.32 | 0.22 | 0.85 | 2.01 |
| Kootenay | BFE | Bonner's Ferry, ID | 1.13 | 0.13 | 0.89 | 1.45 | 1.26 | 0.20 | 0.83 | 2.02 |
| Pend Oreille | ALF | Albeni Falls, ID | 1.26 | 0.14 | 0.96 | 1.68 | 1.65 | 0.30 | 1.02 | 2.97 |
| Flathead | CFM | Columbia Falls, MT | 1.24 | 0.13 | 0.94 | 1.63 | 1.65 | 0.26 | 1.01 | 3.19 |
| Flathead | HGH | Hungry Horse Dam, MT | 1.30 | 0.13 | 1.04 | 1.70 | 1.78 | 0.29 | 1.16 | 3.56 |
| Yakima | KIOW | Yakima, WA | 1.82 | 0.21 | 1.35 | 3.11 | 2.28 | 0.30 | 1.57 | 4.39 |


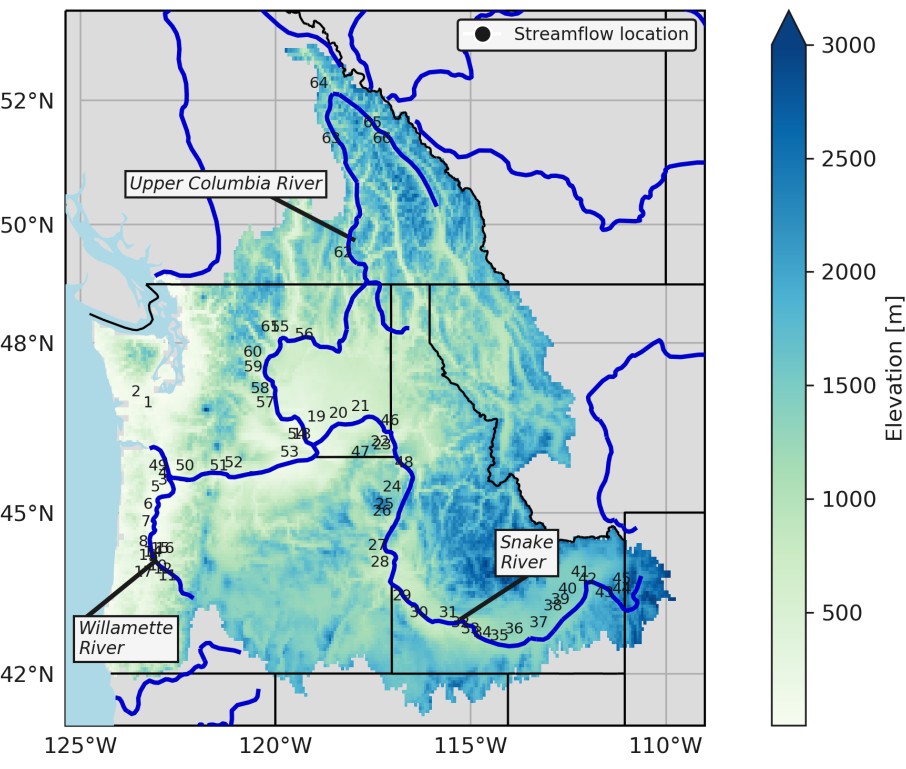

**Figure 1.** Domain of hydrologic simulations used in this paper, with colors indicating elevation of each grid cell, major rivers highlighted in blue, and numbers indicating locations of streamflow points highlighted in Figures 4-9, and Table 1. See Chegwidden et al. (2017, 2019) for all streamflow locations plotted in Figure 3. Digital elevation data are in the public domain, obtained from https://www2.usgs.gov/science/cite-view.php?cite=1530

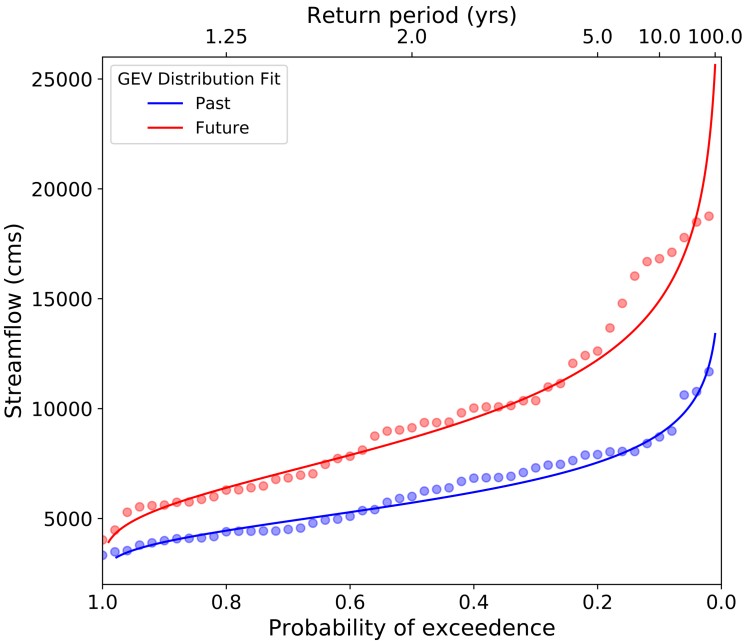

571

**Figure 2.** Generalized Extreme Value fit of annual maximum daily flow from 50 years of simulation using output from one GCM (HadGEM2-ES), one hydrologic model (PRMS), for the Willamette River at Portland. Red and blue dots/ lines indicate the annual values and GEV fit for the 1950-99 'past' and 2050-99 'future' periods.

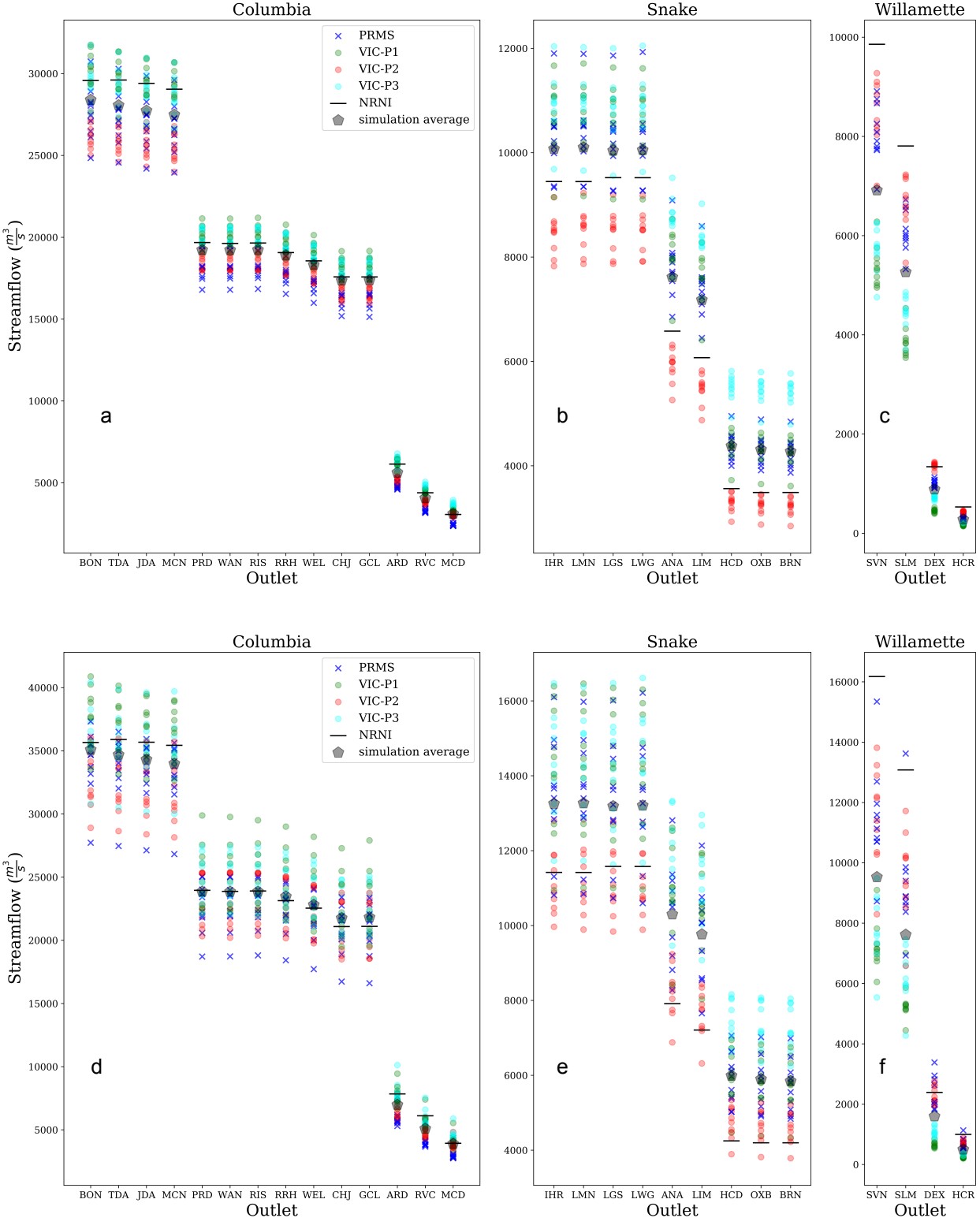

**Figure 3**. Comparison of 10-year (a, b, c) and 100-year (d, e, f) flood magnitudes from the observationally derived NRNI and the 40 climate-hydrologic model simulations, for 1950-2008, for select locations on the rivers as shown.

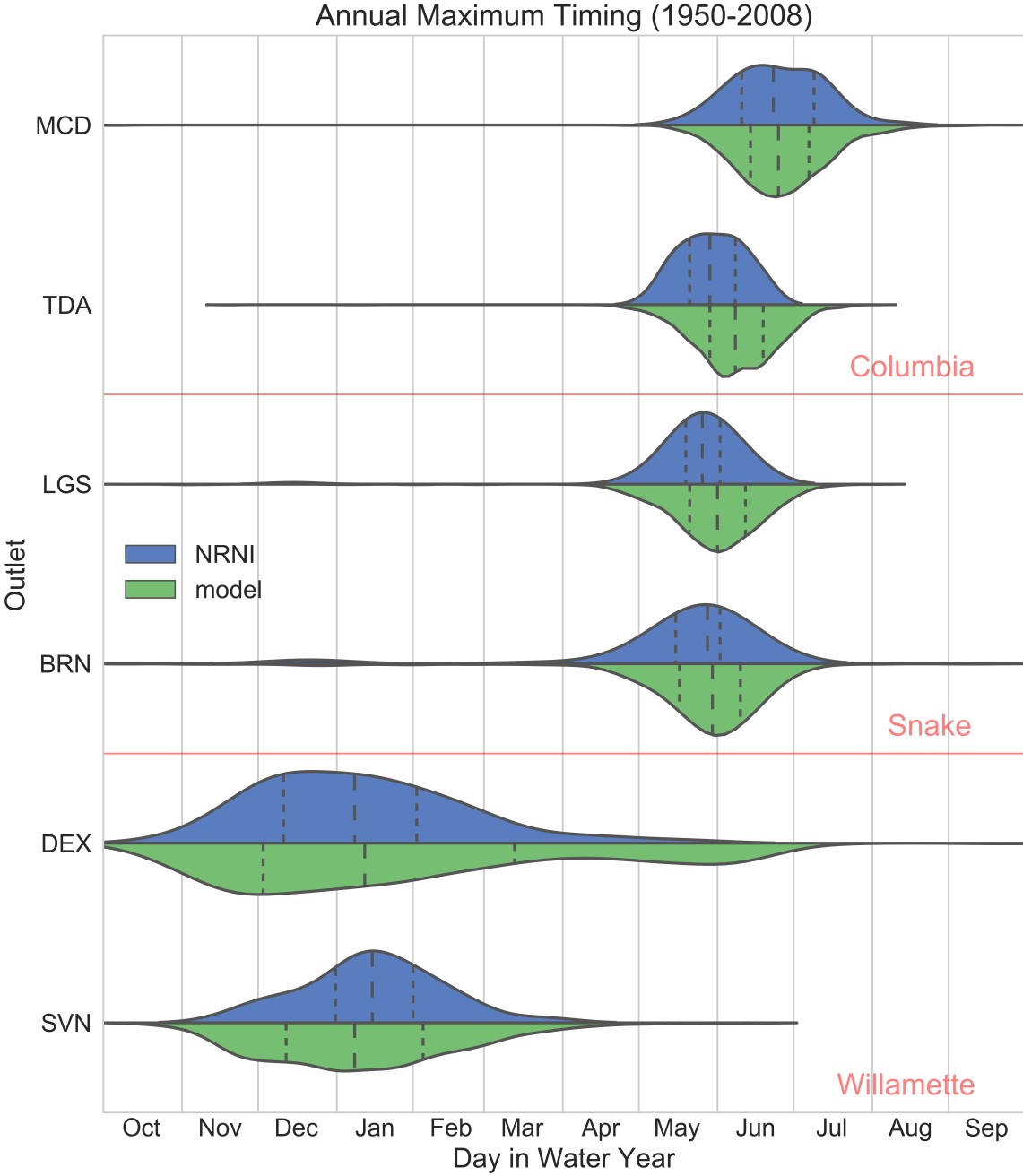

**Figure 4.** Statistical representations of the variation through the water year of the timing of flood events, 1950-2008, for NRNI (blue) and the 40 simulations of 1950-2008 with the climate-hydrology modeling system (green). To create each curve, the dates of the 5 highest streamflows in the period of record are tallied, and the resulting distributions smoothed. Long dashed lines indicate median date, short dashed lines the lowest and highest quartiles. MCD= Mica Dam (upper Columbia), TDA= The Dalles (lower Columbia, between the confluences of the Snake and Willamette), LGS = Little Goose (lower Snake), BRN=Brownlee, SVN=T. W. Sullivan (lower Willamette near Portland), DEX=Dexter (middle fork Willamette).

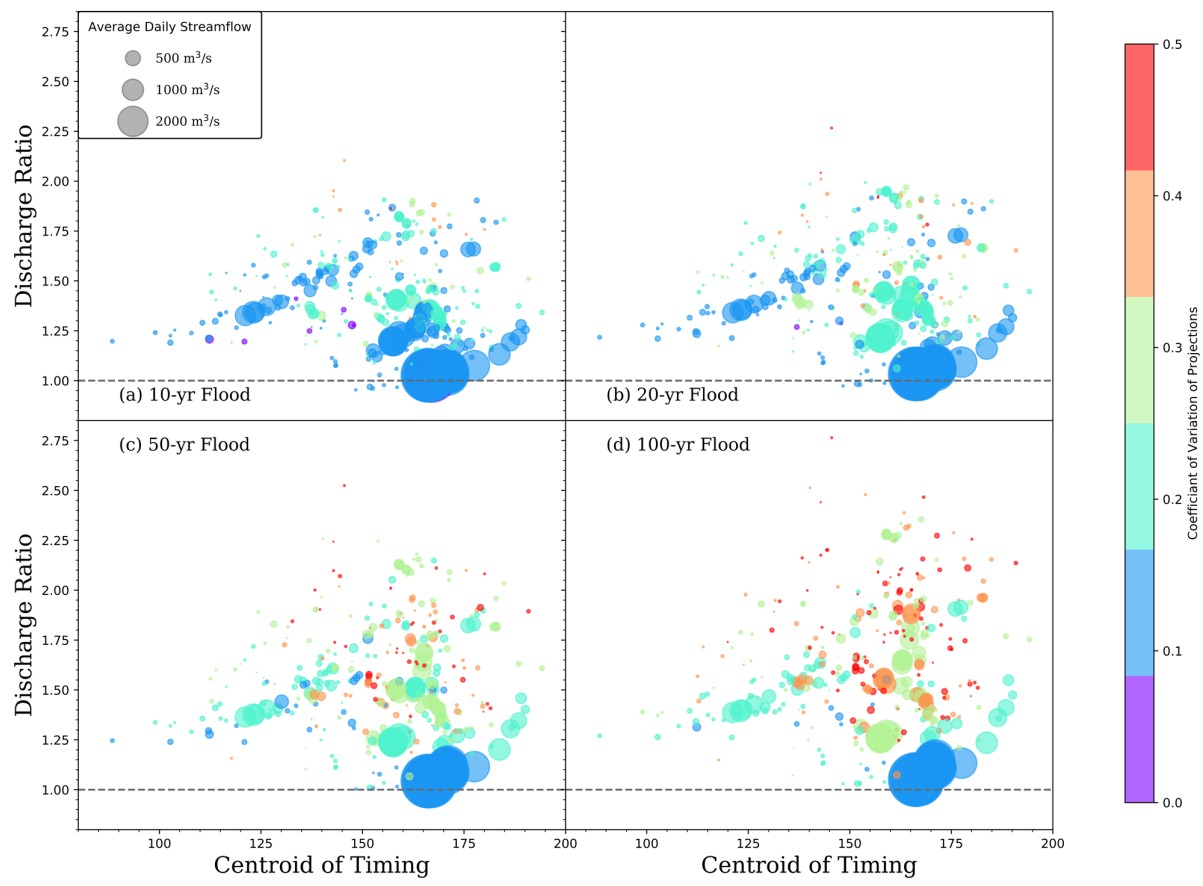

**Figure 5**. Discharge ratios (future:past) versus centroid of timing (day on which 50% of water-year flow has passed, an indicator of snow dominance) for all 396 locations and four return periods. For each location, the average of 40 ensemble member ratios calculated from GEV distribution fitting from 50-year windows for the future (2050-2099) and past (1950-1999) time periods is shown. Points are sized by average daily streamflow and colored by the coefficient of variation of the 40 ratios.

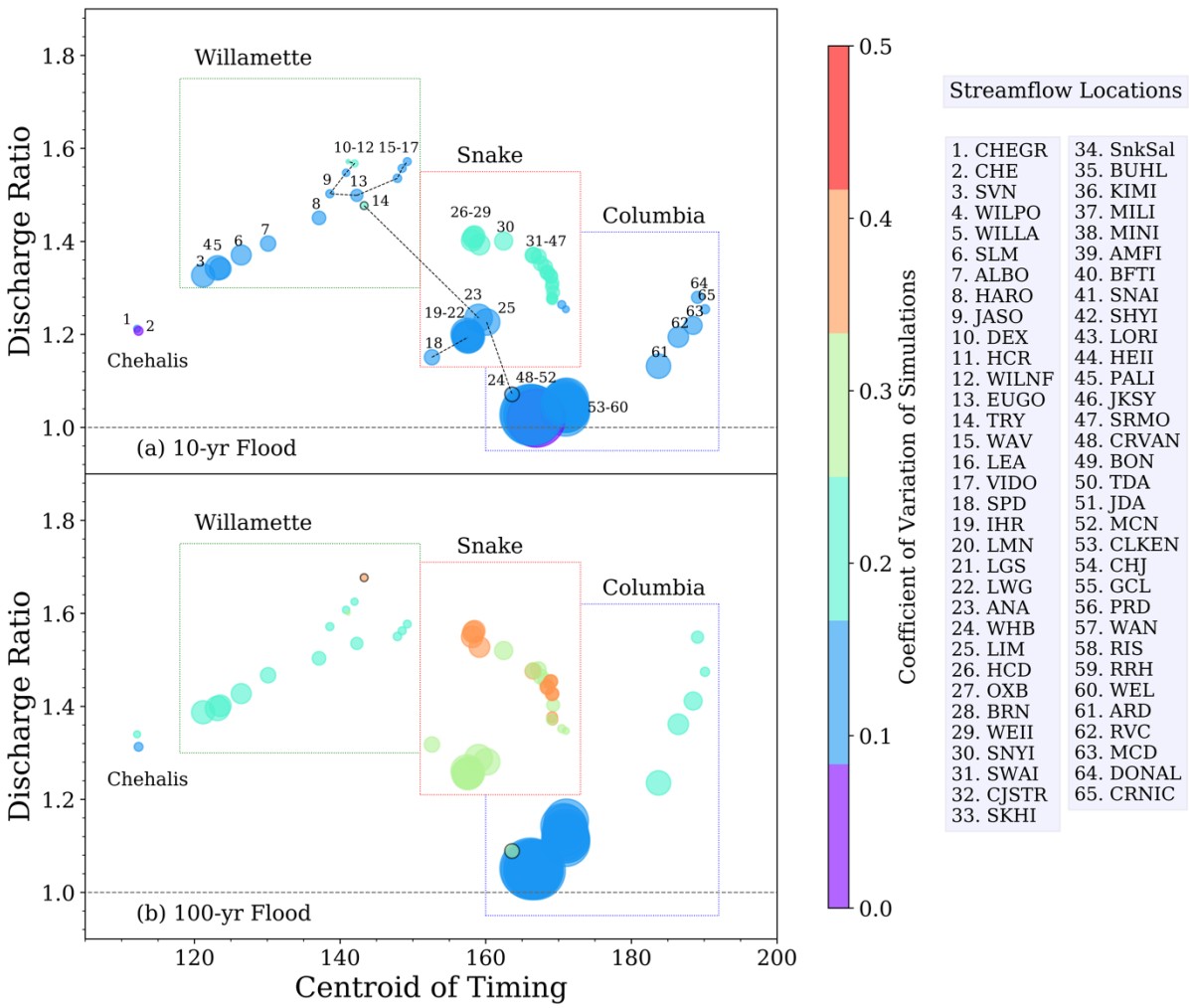

**Figure 6.** As in Figure 5 but only for points on the indicated rivers. Dashed lines indicate tributaries: 9-12 are on the Middle Fork Willamette, 15-17 on the McKenzie; tributaries of the Snake are the Grand Ronde (14), Clearwater (17) and Salmon (24). In the lower panel, the Grand Ronde and Salmon are clearly distinguished by a black circle around their perimeter. Table 1 translates the codes in the legend into named locations and shows the numerical values represented in the figure. As is evident from both snow-dominance and size, locations are ordered downstream to upstream from left to right for each river.

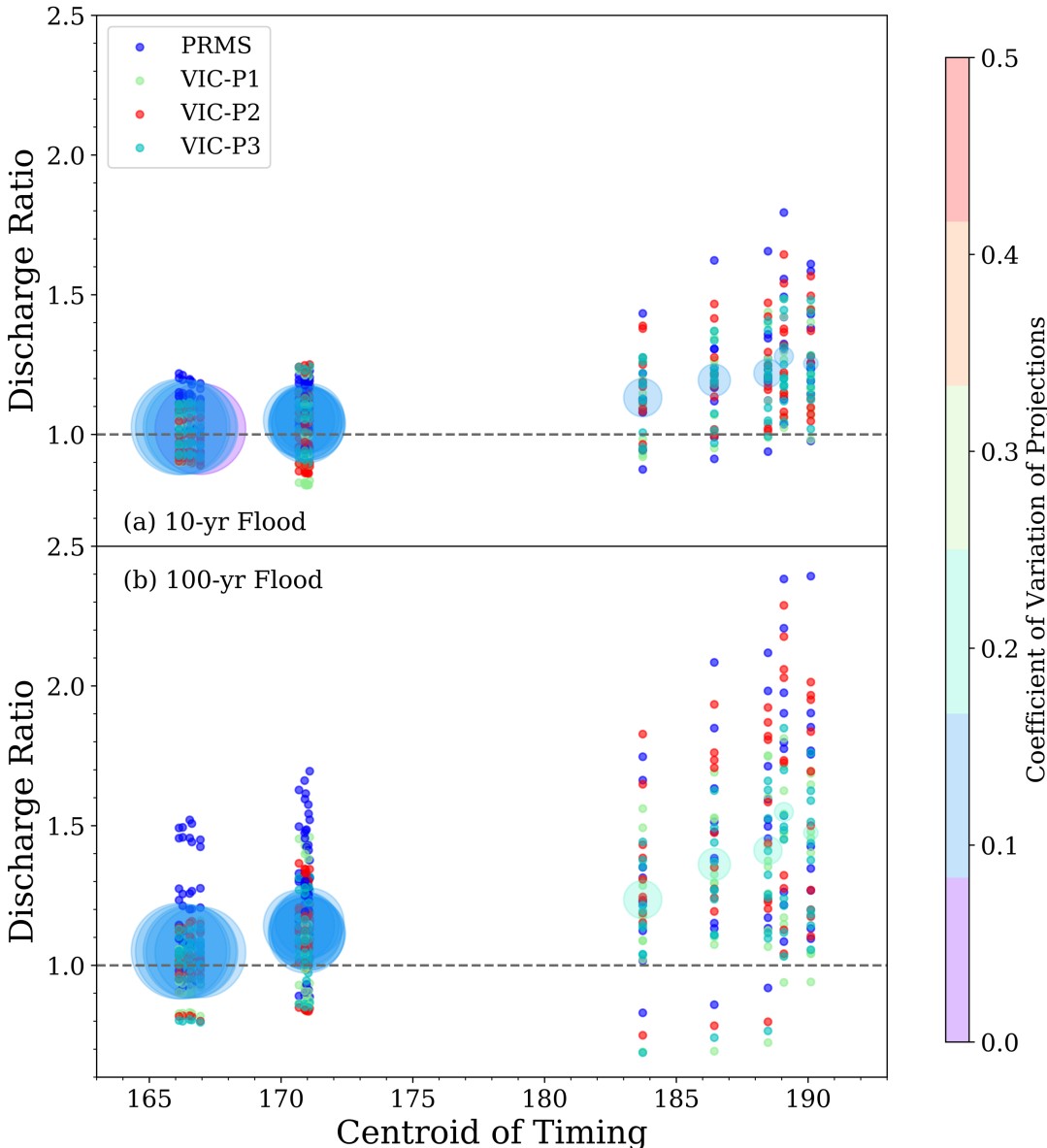

**Figure 7.** Averaged (large circles) and individual ensemble member (small colored circles) discharge ratios for simulated streamflow locations along the mainstem Columbia River for the 10-year (top) and 100-year (bottom) return periods. As shown in the legend, the color of the dots distinguishes results by hydrologic model setup.

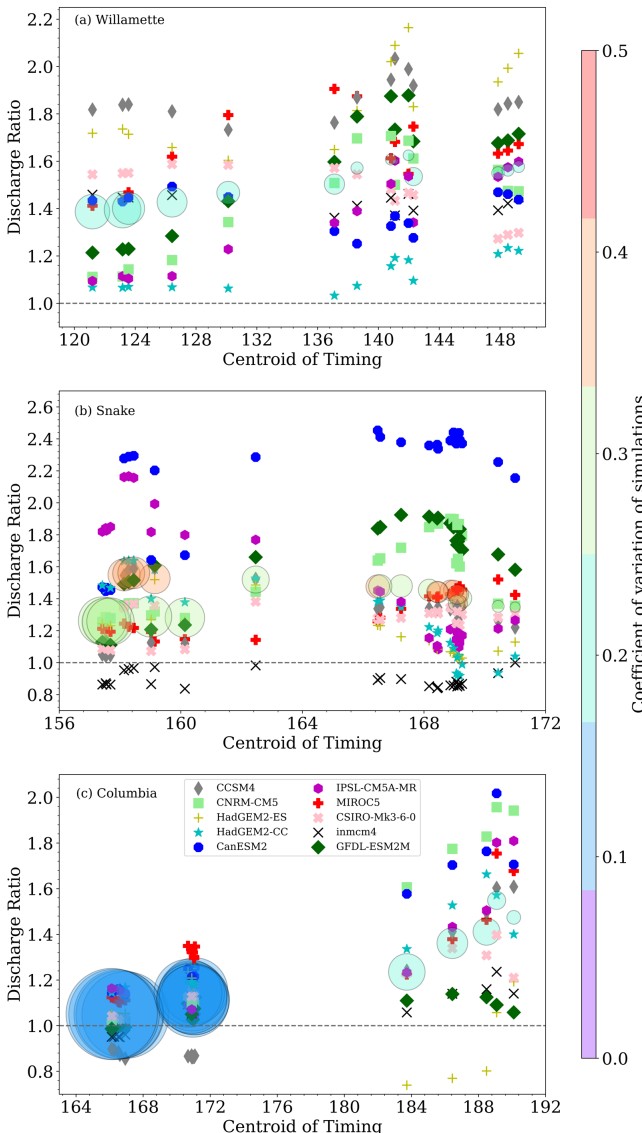

**Figure 8.** Average ratios of all 40 ensemble members (large circles) and the average of 4 hydrologic model re-sults for each GCM (symbols), shown for simulated streamflow locations along the Willamette (top), Snake (middle), and the mainstem Columbia (bottom) for 100-year return periods. GCMs are ordered in the legend by their ranking in Rupp et al. (2017), representing their ability to simulate Northwest climate.

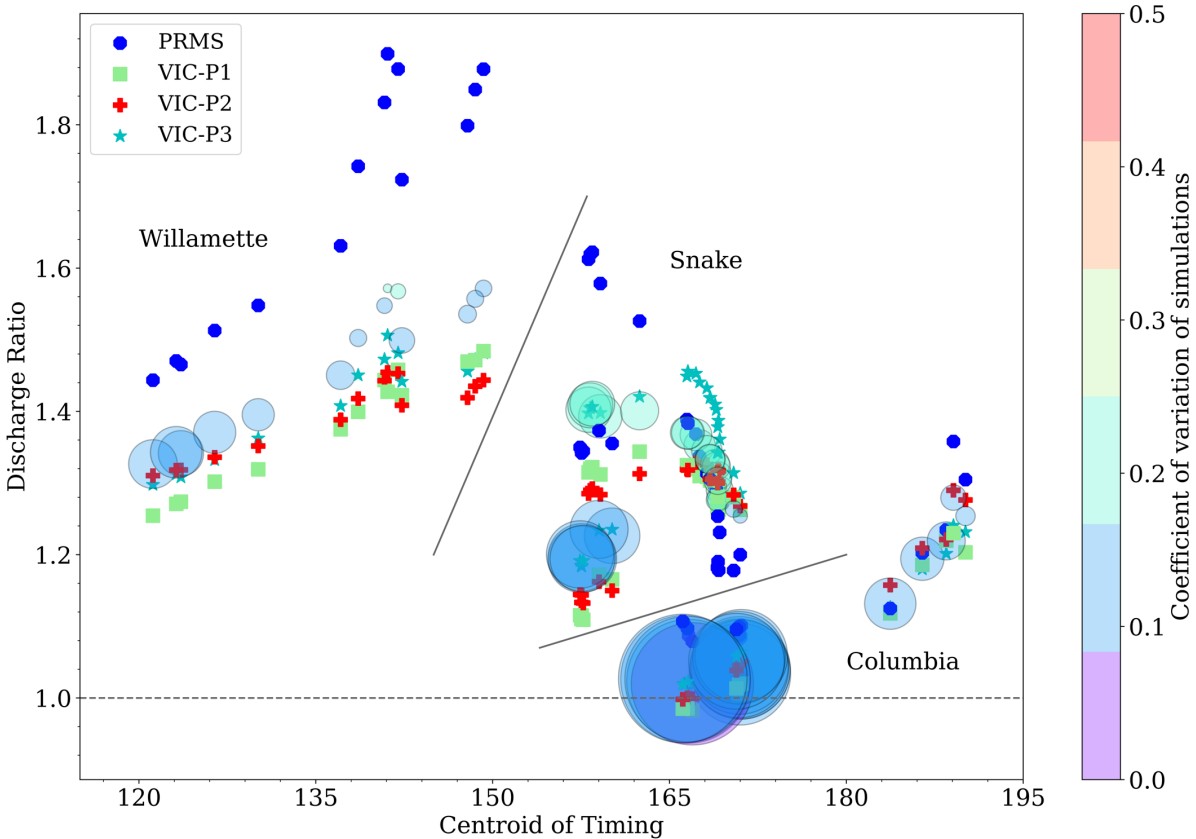

598

599

600

**Figure 9:** as in Figure 8 but averaged by hydrologic model, for 10-year return period, and combined into one panel.

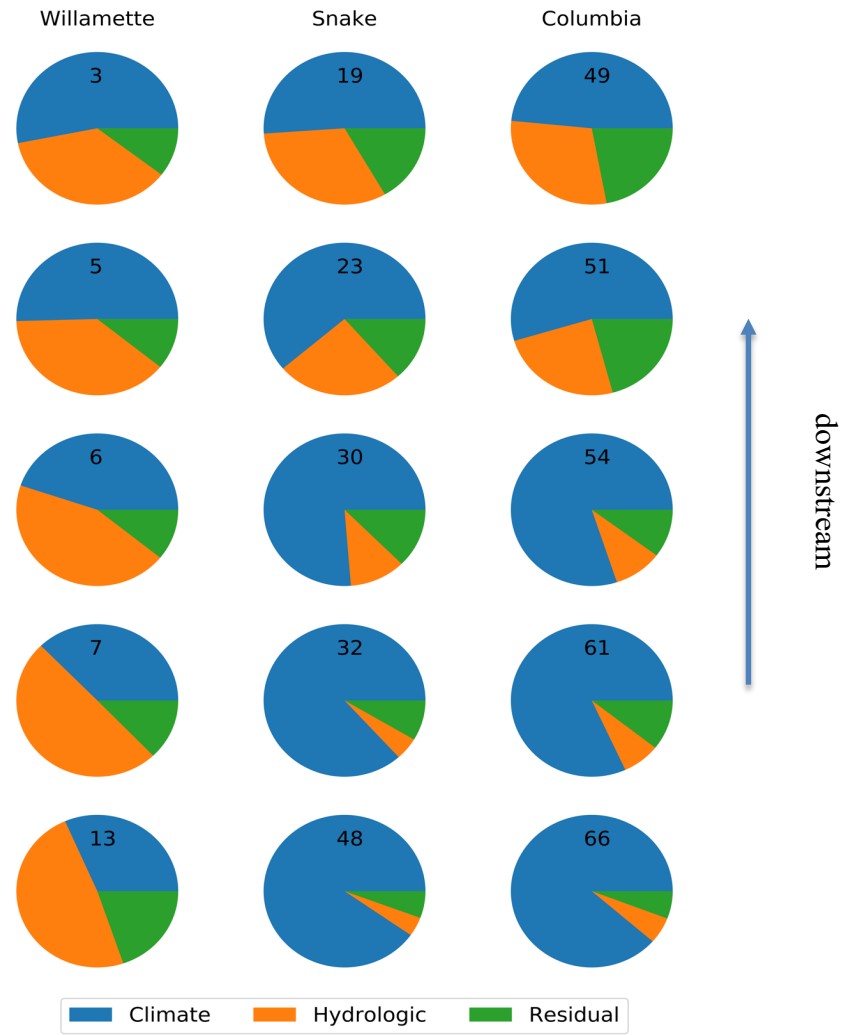

Figure 10. ANOVA results for select locations on the indicated rivers, for climate and hydrologic factors (and the residual). Charts are numbered to correspond with their location in Figure 4, with the most-downstream location at the top. The Snake enters the Columbia after location #54.

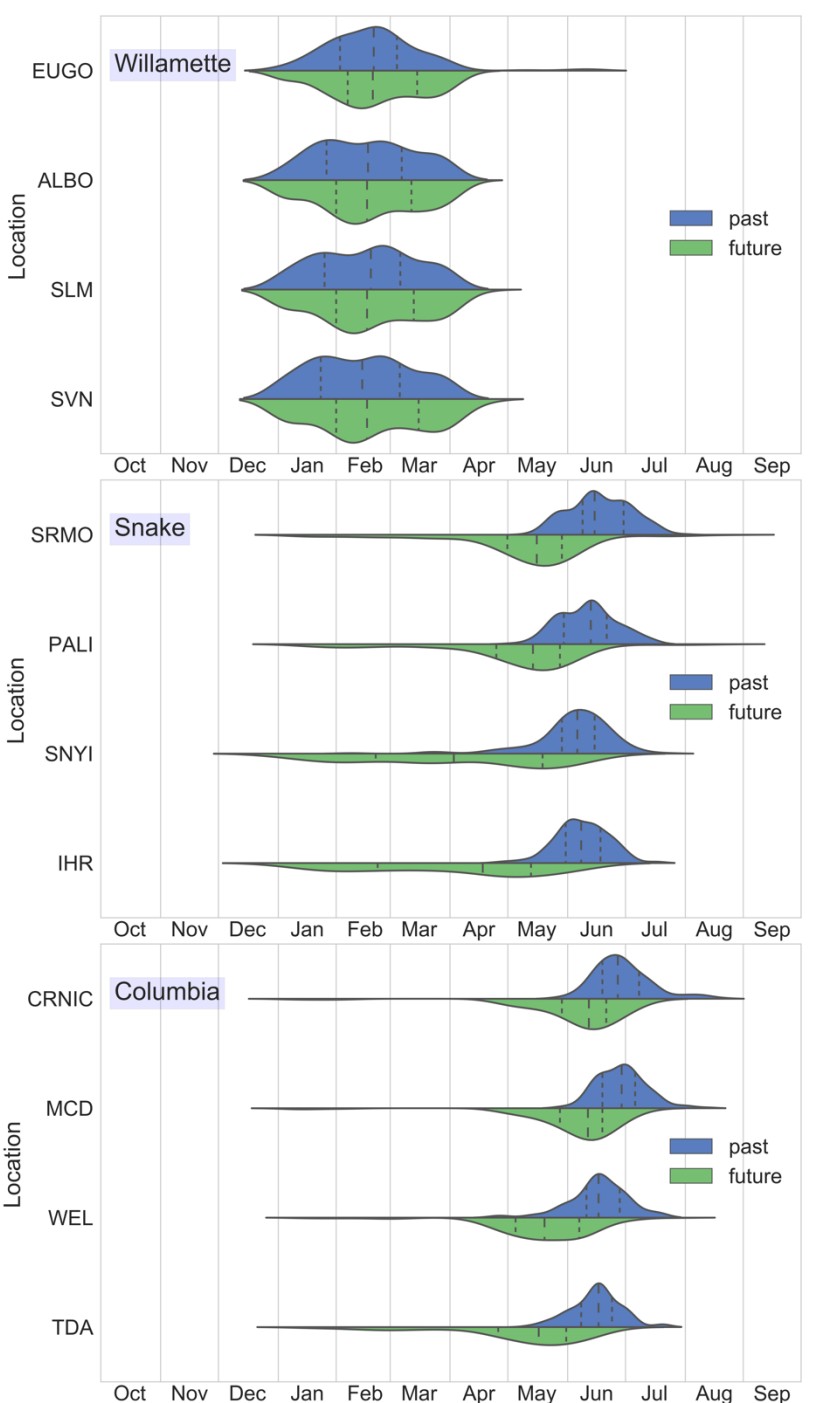

**Figure 11.** Statistical representations of the variation through the water year of the timing of flood events. For each of the 40 simulations, the dates of the 5 highest flows in the 50-year past (blue) and future (green) windows are tallied, and the resulting distributions smoothed. Long dashed lines indicate median date, short dashed lines the lowest and highest quartiles.

606