# Peer review of "Ubiquitous increases in flood magnitude in the Columbia River Basin under climate change"

_Hydrology and Earth System Sciences, 2019_

## Referee Comment (RC1) · Anonymous Referee #1 · 2 Dec 2019

This manuscript contains results from post-processing already existing output from a hydrological modelling system driven by regional climate models. The manuscript is generally well-written, and undoubtedly of interest to local stakeholders. The focus is on interpretation of the results for this particular region, but there appears little or no methodological innovation. In my opinion, the manuscript lacks the originality and significance that would qualify for publication in a leading international journal.
* * *

---

## Short Comment (SC1) · 16 Dec 2019

We thank the reviewer for reading our paper and for praising it as well-written. It is unfortunate that the reviewer says it lacks "originality and significance", which we dispute below, and faults us for providing "little or no methodological innovation".

We first point out that the themes addressed in our paper align with the following guidance from the HESS web site:

"HESS encourages and supports fundamental **and applied research that advances the understanding of hydrological systems, their role in providing water for ecosystems and society**, and the role of the water cycle in the functioning of the Earth system." [emphasis added]

[Figure]

and

"HESS, therefore, aims to serve not only the hydrological science community but all earth and life scientists, **water engineers, and water managers**, who wish to publish original findings on the interactions and feedbacks between the governing processes of the water cycle and processes governing atmospheric circulation and climate, bio-geochemical cycling, dynamics, and resilience of ecosystems and socio-economy.... the study of interactions with human activity of all the processes, budgets, fluxes, and pathways as outlined above, **and the options for influencing them in a sustainable manner, particularly in relation to floods**, droughts, desertification, land degradation, eutrophication, and other aspects of global change." [emphasis added]

We note too that applying methods used previously is not given as grounds for disqualification.
As for the originality of our paper, although there are examples of similar work (cited in our paper), we have not found such a comprehensive study, either for the Columbia Basin or of other large basins, of changing flood risk that accounts for and quantifies key sources of uncertainty (see our ANOVA analysis) and, moreover, describes both the changes in magnitude and seasonality of flood risk and how they change as one travels down a river. Multi-GCM, multi-hydrological model, analyses of changing flood risk across a large area are still very rare (we found one – Thober et al., 2018 – but even they examine only the 1-in-2 year event and they don't explore the hydrological processes that contribute to variability changes in space). If the reviewer is aware of a study that includes the components of our study, we would be grateful to learn of it.

The significance of the paper lies both in its uniqueness and in its generalizability. It is unique in that it provides key numerical input into international treaty negotiations that are currently underway. Many academic papers conclude with a vague admonition to water managers to pay attention to the results. By contrast, most of the authors of

this paper have been deeply involved in developing the key dataset used by the US Entity, and the paper thus has a deep and integral connection to an important policy process. We would be interested to learn whether this critical reviewer has had similar international policy significance arise directly from his/her work.

Second, it is generalizable in that we show how complex the pattern of change (with space and with season) can be in a mixed rain-and-snow basin. Basins of similar size and hydrological response to warming exist on most continents, so our results provide a warning about using that simplistic answers about changing flood risk beyond just the Columbia Basin.

Since the reviewer has offered no suggestions for improving the paper, we await more constructive comments before proposing revisions.

References

Thober, S., Kumar, R., Wanders, N., Marx, A., Pan, M., Rakovec, O., Samaniego, L., Sheffield, J., Wood, E.F. and Zink, M., 2018. Multi-model ensemble projections of European river floods and high flows at 1.5, 2, and 3 degrees global warming. Environmental Research Letters, 13(1), p.014003.

---

## Referee Comment (RC2) · Anonymous Referee #2 · 20 Dec 2019

This paper reports on a new study of the effects of climate change on flood risk in a mountainous river basin that builds substantially on previous published work in the field.

There are several advances in this study over previous work that make it an important contribution to the growing understanding of this problem. In particular, compared to earlier papers, this study uses 1) more recent global climate model simulations, 2) a new statistical downscaling method that better represents daily variability, 3) a suite of hydrologic models to better represent uncertainties. Overall, results are consistent with the previous understanding of flood risk changes in river basins where flooding is snow-melt driven, rain driven, or transitional. Some important difference are seen, however, particularly in the trend for increased flood risk in all regimes, where earlier studies

found decreased risk in some basins. Also, the use of multiple hydrologic models and multiple climate modes provides important insights into the relative importance of uncertainty in climatic and hydrologic processes in different domains.

As with all papers on regional climate change, the paper focuses on a single geographical region. However, the general principles are universal to understanding climate change in regions of complex topography and so, like any regional climate study, is helpful to scientists studying climate change in many other regions worldwide.

Based on these consideration, I find the paper to be a substantive contribution and well deserving of publication subject to revisions outlined below.

I find no major errors in the paper and the relevant literature is properly cited. There are a few issues raised in the paper that I felt were left unresolved. I raise these more as a matter of discussion than as mandatory revisions.

On Page 7, l 193, the authors describe the behavior on the Snake as surprising. I would argue rather that is is the upper Columbia that is surprising. The increase in flood ratio along the Snake is consistent with the shift of snow-dominant to transient basins described in Hamlet and Lettenmaier (2010) and so forth. I found the large flood ratios found in the coldest parts of the Columbia surprising, and this is partly where this study diverges from Tohver et al (2014). I'd like to see more explanation of the Upper Columbia; it's hard to see how the effect of temperature alone could explain this, and there must be an increase in the snowpack at the head waters. The effects of temperature and precipitation could potentially be discerned by comparing changes in discharge timing and intensity (Fig 9) – if intensity changes with out a change in timing, precipitation changes are likely the cause. Results at TDA suggest a longer less abrupt melt season in the future in contrast to CRNIC.

I'd like to see some comparison in the results section to Tohver at al (2014) rather than leave it for the conclusions. Their fig 5 (flipped on y-axis) is essentially the same as your fig 3. Results are essentially the same for rain basins, but you find more

consistent increases for transient and snow basins. Pointing this out here helps show the continuity and newness of these results.

p 8: This section brings a lot of fresh insight, and I'd like to see a little more. In particular, given the interesting result of high flood rations in the Upper Columbia, it would be interesting to understand what's different about models like cnrm-cm5 and canesm2 that show the strongest result.

l232: Also, for the snow dominant basins, much of the uncertainty depends on how much snowpack is simulated for the future and how it melts, which are related to temperature and precipitation. Putting it into the river is relatively easy.

Minor quibble: At l232 you say "the role of climate grows more important and the role of hydrologic variability less important." The word "role" suggests that in the real world, changes in climate and hydrology have differing effects. Really, it's just the uncertainty in modeling these that map onto uncertainty in the results differently depending on location. So

In the conclusions, you make comparisons to Salathé et al (2014), and attribute a lot of the difference to the single scenario used there. Another factor is that the RCM is going to have a lot shorter spatial coherence (ie lack of temporal correlation across small distances) in the precipitation. So, whether it is snow or rain, for a single realization, there will be a few stations that buck the larger (deterministic) trend by chance. A large ensemble would reduce that effect.

---

## Referee Comment (RC3) · Anonymous Referee #2 · 20 Dec 2019

In perfunctory comments, reviewer #1 suggests that a regional study is not of much interest to international readers outside the domain of the study. I find this an absurd assertion, one that would make all regional climate research unpublishable. Indeed, in the most recent issue of HESS, the majority of articles focus on small regional domains (just from titles: Iberian Penninsula, Sao Paolo, Lake Erken, Spain, China, Athabasca River, Nile Delta). https://www.hydrol-earth-syst-sci.net/23/issue12.html

Secondly, the implication that publishable science articles must be based on methodological innovation rather than the analysis of physical processes, revealed by methodological innovations, is baffling.

[Figure]

474, 2019.

---

## Referee Comment (RC4) · Anonymous Referee #3 · 15 Jan 2020

This paper performs a fairly thorough assessment of change in the Columbia river basin as assessed using a combination of GCM and hydrologic models. The assessment is quite good and the authors have done a lot of work for which they are to be commended.

I have concerns though about the novelty of this work. My first concern is the use of GCMs and not RCMs for which CMIP5 simulations are now available over most of North America. My second concern is the use of an analogue based downscaling approach which may be compromised in its ability to represent unseen extremes (I say this without reading the cited paper though but was surprised this was used and not the more fancy downscaling plus bias correction alternatives that are around nowadays). In general, while this study is well presented, I feel there is little I can use in terms

of methods for my own applications which may pertain to a different catchment. My recommendation is to shift this paper to a more applied journal that may be more inviting of regional contributions where this will be better appreciated I believe.

Alternately the authors must try to capture some novel question in their analysis that may shed light on processes elsewhere. For instance, a significant portion of the flow in the Columbia comes because of melt. Additionally, it is well known GCM simulations are not very reliable in the context of precipitation. Is there a research question in how one could downscale snow and rain using GCMs in a way snowpack dynamics for the current climate period are well represented? Additionally, how this downscaling would comapre with the higher spatial scale simulations from RCMs over the study region. There may be other questions too that could be of interest. Given the work the authors have already done, I urge them to identify such questions and change their presentation to addressing these instead of reporting overall changes in the basin.

---

## Referee Comment (RC5) · Anonymous Referee #4 · 5 Feb 2020

generated using precipitation projections by Global Circulation Models (GCM) downscaled to daily values used in combination with four conceptual hydrologic models for the period 1950-2099. The authors compared the flood statistics for period 1950-1999 with period 2050-2099 and found that there was a general increase in the daily maximum flood ratio between the two periods. The flood statistics were based on 40 model ensembles consisting of 4 hydrologic models and 10 GCM projections. Along with flood ratio, the authors also look at changes to flood timing in the basin and suggest that reservoir rules should be revised to account for changes in a future climate. While the topic and the premise of this paper is very important and relevant to HESS and in the current context of climate change, and the fact that the Columbia River system is critical to the western USA, the paper needs substantial revisions to

both content, writing and presentation prior to accepting for publication. Following are the three major comments/points/questions that should be addressed in the revisions. Additional minor comments are listed in the table below. 1. How does the daily rainfall and resultant flood statistics compare with historic data? How well does the GCM downscaling match up and how well does the 4 hydrologic model generated flood event compare with the historic data for magnitude and timing. As mentioned in the paper, the previous work was based on annual and monthly flood statistics and the change in modelling timesteps require some sort of validation. Acknowledging that this basin has a lot of flow control, even the comparison of the upper reaches of the river system might be sufficient to gain some level of comparison on how well the four hydrologic model simulations compare with historic data. 2. As addressed in the conclusion of this paper, the question on how much of a contribution does the PRMS model results have on the increase in flood ratios needs to be addressed in some way as this paper suggests an increase in flood risk which is different to many other studies. A possible way that can be considered might be, were PRMS flood predictions higher for both periods of comparison? How do the result change if the results from the PRMS models are not considered? 3. The introduction needs to present the objective and the importance of this paper better than a single line at the end. The introduction should present the case for why this work is important. Also, the methods section needs more detail on how things are different from the previous work from which this paper was derived. # Line #(s) Comment 1 43 Indicate the overall area of the basins that are part of this study 2 53 Need a citation or some additional support on this sentence. 3 63 This is an important point which relates to a lot of this study and should be explained better to show relevance and why this stud can be useful. 4 95 Similar to above comment, the gap that this paper looks to explore and study is not made clear. Why is this study important and how does it differ from the study that this work is based on? What is the primary objective of this paper? 5 128-135 At are the 4 models? Descriptions provided for only 2 of the models used. What level of conceptualization and under lying assumptions rule these models? 6

161-179 Language and sentence structure should be edited. Vague language such as 'sometimes' 'more or less' 'just below' should be avoided 7 181 The change in flood magnitude along the river should also include some discussion on tributaries contributing flow 8 191 Correct me if I am wrong, but Isn't base flow accounted for in the extreme value analysis? 9 200 delete this sentence 10 203-205 Move to start of this section 11 299-311 This paragraph is good

Please also note the supplement to this comment:
https://www.hydrol-earth-syst-sci-discuss.net/hess-2019-474/hess-2019-474-RC5-supplement.pdf

---

## Short Comment (SC2) · 14 Feb 2020

Laura Queen

queenl@oregonstate.edu

We are grateful to Reviewer 3 for taking the time to perform this review and for noting the effort required to conduct this thorough assessment. We address below the reviewer's concerns.

"My first concern is the use of GCMs and not RCMs."

We have several responses to this concern.

1) Large ensembles of RCMs are rare. The 12-member NARCCAP ensemble (6 RCMs, 4 GCMs), completed a decade ago, remains the largest, but has a spatial resolution of only 50km. CORDEX North America now has a comparable-size ensemble, but mostly still at 50 km (some at 0.22°), and was not available in such large numbers

when we began our hydrologic simulations. At such spatial resolutions, RCMs would still have to be further downscaled and bias corrected to use in our hydrologic models (∼6km spatial resolution). Thus, RCMs are not necessarily a vastly better solution.

2) RCMs certainly have their place in such work and were used in some previous studies noted in our paper. But this dataset was developed in order to sample a larger climate space than is possible with RCMs, which must be driven by GCMs anyway and were too resource intensive to run to generate the 40 different climate scenarios used here.

3) Our ANOVA analysis (Figure 8) shows that the climate scenarios contribute a majority of the variation among results for most of the basin. Consequently, it is of great importance to sample the climate scenarios. Using RCMs would constrain us to a much smaller range of climate scenarios.

4) The flood events for the Snake and Columbia have a significant snowmelt component. As a result, the value of simulating hydrological processes well probably exceeds the additional value of RCMs in simulating daily rainfall compared to the MACA approach which links large-scale flow to local processes through a constructed analogs approach.

5) The mere fact that an alternative approach exists should not mean that the current approach, which has substantial backing in the literature, is rejected. Using RCMs would be an entirely different study, with (as noted above) its own weaknesses.

"My second concern is the use of an analogue based downscaling approach which may be compromised in its ability to represent unseen extremes"

The MACA dataset has been in wide use for nearly a decade while undergoing improvements since Abatzoglou and Brown (2011), and although other approaches exist, the most recent improvement on MACA (LOCA, the approach used in the most recent National Climate Assessment) also uses constructed analogs. In short, this approach

is still the benchmark. Points 4 and 5 above are relevant here too. Alder and Hostetler (2018) compared MACA and LOCA and found they gave similar results in hydrologic modeling.

"Alternately the authors must try to capture some novel question in their analysis that may shed light on processes elsewhere. For instance, a significant portion of the flow in the Columbia comes because of melt. Additionally, it is well known GCM simulations are not very reliable in the context of precipitation. Is there a research question in how one could downscale snow and rain using GCMs in a way snowpack dynamics for the current climate period are well represented? Additionally, how this downscaling would comapre [sic] with the higher spatial scale simulations from RCMs over the study region. There may be other questions too that could be of interest. Given the work the authors have already done, I urge them to identify such questions and change their presentation to addressing these instead of reporting overall changes in the basin."

The mechanisms of flooding in the upper Columbia and elsewhere are a key question arising from this work, and we agree with the reviewer that further investigation is merited. While beyond the scope of this paper, we have other papers in process that address some relevant questions: Chegwidden et al (in review) look at the processes that contribute to sensitivity of flood magnitude to changes in climate, and they assess how climate change will alter high streamflow events by both changing the prevalence of the flood generating process and the magnitude of differently generated floods. They present an analysis of changes in high streamflow events, classifying the events according to their underlying mechanisms, and compare how the different kinds of high flows respond to changes in climate at the annual scale. They find that snow will play a diminished role in generating high flows in the future. High flow events will switch to being caused by precipitation events, which they find are also more sensitive than snowmelt-driven events to increases in precipitation.

We contend that our approach is novel: the reviewer has not pointed us toward, nor could we find, a paper in the literature that uses such a large climate-hydrological ensemble to comprehensively characterize the changes in flood magnitude over a basin, let alone while systematically presenting the dependence of results on climate scenario, location/hydrological characteristics, and other factors. Our ANOVA results, and plots distinguishing the variation across climate scenarios and hydrology scenarios, are all unique in the literature as far as we are aware. Moreover, the very purpose of this dataset – to inform international treaty negotiations – sets it apart from standard academic research. The reviewer's other suggestions to investigate snowpack dynamics or to use RCMs, are considerably beyond the scope of this paper.

In conclusion, we contend that our dataset and our approach are sufficiently state-of-the-science to merit publication, that HESS publishes papers of similar novelty and geographic focus, that RCMs are one (but not the only) acceptable tool for scientific studies such as ours, and that in the absence of similar work by others, which Reviewer 3 has not furnished, our work is novel.

References

Abatzoglou, J. T., & Brown, T. J. (2012). A comparison of statistical downscaling methods suited for wildfire applications. International Journal of Climatology, 32(5), 772-780.

Alder J R and Hostetler S W 2019 The Dependence of Hydroclimate Projections in Snow-Dominated Regions of the Western United States on the Choice of Statistically Downscaled Climate Data Water Resour. Res. 55 2279–300

Chegwidden, O., D. Rupp, and B. Nijssen, 2020: Upstream processes determine flood response to climate change. Environmental Research Letters, in review.

---

## Author Comment (AC1) · 26 Feb 2020

We are grateful to Reviewer 4 for a constructive and insightful review, and for describing the paper as important and relevant to HESS. Below we outline our plans for heeding the excellent points raised in this review.

1. Major revisions to the paper would be required to meet this quite appropriate and thoughtful recommended improvement. We suggest an analysis of the historical representation of annual maximum flows within the hydrologic modeling setup. As a comparison, we will use a set of streamflows called No Reservoirs No Irrigation (NRNI; RMJOC 2017). Developed by federal agencies to support practical analysis, the NRNI dataset exists at ∼190 sites across the Columbia River Basin, and are adjusted to correct for reservoir management and the diversions and evaporation associated with both the reservoirs and with irrigated agriculture. This dataset is suitable for comparisons with our modeling setup, and we plan to use all the NRNI locations in this analysis, paying particular attention to locations along the Willamette, Columbia, and Snake and in basins with minimal flow control. We will compare the annual maximum daily flow statistics (return period curves as in Figure 2, without GEV fits; and twin-violin plots as in Figure 9) for 1960-2008 from the observed and simulated records. These historical performance results will inform our interpretation of the projected changes in annual maximum daily flow. We would select an appropriate metric like the 90th percentile of annual maximum daily flows, and compare the historical simulations.

2. The above analysis may also help to assess the quality of the PRMS 20th century simulations and provide guidance about whether the PRMS results should be excluded from the ensemble analysis. From Figure 7, it is clear that such a decision would change the results in some places like the lower Columbia and the Snake.

3 The points raised in the table are good recommendations for clarifying and expanding the text and we will address them all in our revisions.

River Management Joint Operating Committee (RMJOC). (2017). NRNI Flows 1929-2008 Corrected 04-2017. Bonneville Power Administration. Retrieved from https://www.bpa.gov/p/Power-Products/Historical-Streamflow-Data/Pages/No-Regulation-No-Irrigation-Data.aspx

Please also note the supplement to this comment:
https://www.hydrol-earth-syst-sci-discuss.net/hess-2019-474/hess-2019-474-AC1-supplement.pdf

---

## Author Comment (AC2) · 26 Feb 2020

We find reviewer 2's suggestions excellent and will follow them when revising the paper.

---

## Author Comment (AC3) · 26 Feb 2020

We appreciate and agree with Reviewer 2's views.
* * *

---

## Author Comment (AC4) · 26 Feb 2020

See response posted by lead author Laura Queen. System requires me to respond.

---

## Author Response (AR1)

Revised responses to reviewers - Text in blue is added since our online responses

——————————- REVIEWER 1 ————————————

We thank Reviewer 1 for reading our paper and for praising it as well-written. It is unfortunate that the reviewer says it lacks "originality and significance", which we argue against below, and faults us for providing "little or no methodological innovation".

We first point out that the themes addressed in our paper align with the following guidance from the HESS web site:

> HESS encourages and supports fundamental and **applied research that advances the understanding of hydrological systems, their role in providing water for ecosystems and society**, and the role of the water cycle in the functioning of the Earth system. [*emphasis added*]

and

> HESS, therefore, aims to serve not only the hydrological science community but all earth and life scientists, water engineers, and water managers, who wish to publish original findings on the interactions and feedbacks between the governing processes of the water cycle and processes governing atmospheric circulation and climate, bio-geochemical cycling, dynamics, and resilience of ecosystems and socio-economy.
> [.....]
> 3. the study of interactions with human activity of all the processes, budgets, fluxes, and pathways as outlined above, and the options for influencing them in a sustainable manner, particularly in relation to floods, droughts, desertification, land degradation, eutrophication, and other aspects of global change. [*emphasis added*]

We next note that applying methods use previously is not given as a criterion for disqualification,

As for the originality of our paper, although there are examples of similar work (cited in our paper), we have not found such a comprehensive study, either for the Columbia Basin or of other large basins, of **changing flood risk** that accounts for and quantifies key sources of uncertainty (see our ANOVA analysis) and, moreover, describes both the changes in magnitude and seasonality of flood risk and how they change as one travels down a river.   Multi-GCM, multi-hydrological model, analyses of changing flood risk across a large area are still very rare (we found one – Thober et al., 2018 – but even they examine only the 1-in-2 year event and they don't explore the hydrological processes that  contribute to variability changes in space) If the reviewer is aware of a study that includes the components of our study, we would be grateful to learn of it.

The significance of the paper lies both in its uniqueness and in its generalizability. It is unique in that it provides key numerical input into international treaty negotiations that are currently underway. Many academic papers conclude with a vague admonition to water managers to pay attention to the results. By contrast, most of the authors of this paper have been deeply involved in

developing the key dataset used by the US Entity, and the paper thus has a deep and integral connection to an important policy process. Second, it is generalizable in that we show how complex the pattern of change (with space and with season) can be in a mixed rain-and-snow basin. Basins of similar size and hydrological response to warming exist on most continents, so our results provide a warning about using that simplistic answers about changing flood risk beyond just the Columbia Basin. We have added some of this response to the text.

**References**

Thober, S., Kumar, R., Wanders, N., Marx, A., Pan, M., Rakovec, O., Samaniego, L., Sheffield, J., Wood, E.F. and Zink, M., 2018. Multi-model ensemble projections of European river floods and high flows at 1.5, 2, and 3 degrees global warming. *Environmental Research Letters*, *13*(1), p. 014003.

————————————— REVIEWER 2 ——————————————————

We are grateful to Reviewer 2 for a particularly thoughtful review and also for responding to Reviewer 1.

On Page 7, l 193, the authors describe the behavior on the Snake as surprising. I would argue rather that is is the upper Columbia that is surprising. The increase in flood ratio along the Snake is consistent with the shift of snow-dominant to transient basins described in Hamlet and Lettenmaier (2010) and so forth. I found the large flood ratios found in the coldest parts of the Columbia surprising, and this is partly where this study diverges from Tohver et al (2014). I'd like to see more explanation of the Upper Columbia; it's hard to see how the effect of temperature alone could explain this, and there must be an increase in the snowpack at the head waters. The effects of temperature and precipitation could potentially be discerned by comparing changes in discharge timing and intensity (Fig 9) – if intensity changes with out a change in timing, precipitation changes are likely the cause. Results at TDA suggest a longer less abrupt melt season in the future in contrast to CRNIC.

We made some changes to the text but a full analysis of the Upper Columbia will require work beyond the scope of this paper.

I'd like to see some comparison in the results section to Tohver at al (2014) rather than leave it for the conclusions. Their fig 5 (flipped on y-axis) is essentially the same as your fig 3. Results are essentially the same for rain basins, but you find more consistent increases for transient and snow basins. Pointing this out here helps show the continuity and newness of these results.

We added some text to the results section as suggested.

p 8: This section brings a lot of fresh insight, and I'd like to see a little more. In particular, given the interesting result of high flood rations [*sic*] in the Upper Columbia, it would be interesting to understand what's different about models like cnrm-cm5 and canesm2 that show the strongest result.

Analysis of the sort suggested would require detailed analysis of the individual GCM runs, and is beyond the scope of this paper.

At l232 you say "the role of climate grows more important and the role of hydrologic variability less important." The word "role" suggests that in the real world, changes in climate and hydrology have differing effects. Really, it's just the uncertainty in modeling these that map onto uncertainty in the results differently depending on location. So

We changed the text as suggested.

In the conclusions, you make comparisons to Salathé et al (2014), and attribute a lot of the difference to the single scenario used there. Another factor is that the RCM is going to have a lot shorter spatial coherence (ie lack of temporal correlation across small distances) in the precipitation. So, whether it is snow or rain, for a single realization, there will be a few stations that buck the larger (deterministic) trend by chance. A large ensemble would reduce that effect.

Great point. We changed the text accordingly.

———————————————— REVIEWER 3 ————————————————

We are grateful to Reviewer 3 for taking the time to perform this review and for noting the effort required to conduct this thorough assessment. We address below the reviewer's concerns.

"My first concern is the use of GCMs and not RCMs."

We have several responses to this concern. We have added substantial text in the manuscript, including all the key elements of our responses below, to address this concern.

1) RCMs certainly have their place in such work and were used in some previous studies noted in our paper. But this dataset was developed in order to sample a larger climate space than is possible with RCMs, which must be driven by GCMs anyway and are too expensive to run to generate the 40 different climate scenarios used here. As far as we are aware, the most comprehensive published dataset of RCM-GCM combinations is the NARCCAP ensemble which is now nearly a decade old and was conducted using just 6 RCMs, all at 50km spatial resolution.

2) Our ANOVA analysis (Figure 8) shows that the climate scenarios contribute a majority of the variation among results for most of the basin. Consequently, it is of great importance to sample the climate scenarios. Using RCMs would constrain us to a much smaller range of climate scenarios.

3) RCMs are still at too coarse a resolution to use in our simulations, and thus would have to be further downscaled and most likely bias corrected.

4) Since the flood events for the Snake and Columbia have a significant snowmelt component, the value of getting the hydrological processes right probably exceeds the additional value of RCMs in simulating daily rainfall vs the MACA approach for linking large-scale flow to local processes through the constructed analogs approach.

5) The mere fact that an alternative approach exists should not mean that the current approach, which has substantial backing in the literature, is rejected. Using RCMs would be an entirely different study, with (as noted above) its own weaknesses.

> "My second concern is the use of an analogue based downscaling approach which may be compromised in its ability to represent unseen extremes"

The MACA dataset has been in wide use for nearly a decade while undergoing improvements, and although other approaches exist, the most recent improvement on MACA (LOCA) also uses constructed analogs. In short, this approach is still the benchmark. Points 4 and 5 are relevant here too. We have added a new verification section, 2.3, to show the performance of the modeling setup.

> Alternately the authors must try to capture some novel question in their analysis that may shed light on processes elsewhere. For instance, a significant portion of the flow in the Columbia comes because of melt. Additionally, it is well known GCM simulations are not very reliable in the context of precipitation. Is there a research question in how one could downscale snow and rain using GCMs in a way snowpack dynamics for the current climate period are well represented? Additionally, how this downscaling would comapre [*sic*] with the higher spatial scale simulations from RCMs over the study region. There may be other questions too that could be of interest. Given the work the authors have already done, I urge them to identify such questions and change their presentation to addressing these instead of reporting overall changes in the basin.

We contend that our approach is novel – the reviewer has not pointed us toward, nor could we find, a paper in the literature that uses such a large climate-hydrological ensemble to characterize the changes in flood magnitude over a basin, let alone while systematically presenting the dependence of results on climate scenario, location/hydrological characteristics, and other factors. Our ANOVA results, and plots distinguishing the variation across climate scenarios and hydrology scenarios, are all unique in the literature as far as we are aware. Moreover, the very purpose of this dataset – to inform international treaty negotiations – sets it apart from standard academic research. The reviewer's other suggestions to investigate snowpack dynamics or to use RCMs, are considerably beyond the scope of this paper.

In conclusion, we contend that our dataset and our approach are sufficiently state-of-the-science to merit publication, that HESS publishes papers of similar novelty and geographic focus, and that the reviewer has not provided enough evidence either that RCMs are the only acceptable tool for scientific studies such as ours nor that others have effectively done the same kind of study.

———————————————— REVIEWER 4 ————————————————

We are grateful to Reviewer 4 for a constructive and insightful review, and for describing the paper as important and relevant to HESS. Below we outline our plans for heeding the excellent points raised in this review.

1. How does the daily rainfall and resultant flood statistics compare with historic data? How well does the GCM downscaling match up and how well does the 4 hydrologic model generated flood event

compare with the historic data for magnitude and timing. As mentioned in the paper, the previous work was based on annual and monthly flood statistics and the change in modelling timesteps require some sort of validation. Acknowledging that this basin has a lot of flow control, even the comparison of the upper reaches of the river system might be sufficient to gain some level of comparison on how well the four hydrologic model simulations compare with historic data.

We added a new section 2.3 to summarize new analysis we performed in which we compare our large ensemble (hydrologic models driven by free-running GCMs, not observed meteorology) with 'observed' streamflows from a dataset called No Reservoirs No Irrigation (NRNI; RMJOC 2017). We compared return period curves as in Figure 2, without GEV fits, and twin-violin plots as in Figure 9.

2. As addressed in the conclusion of this paper, the question on how much of a contribution does the PRMS model results have on the increase in flood ratios needs to be addressed in some way as this paper suggests an increase in flood risk which is different to many other studies. A possible way that can be considered might be, were PRMS flood predictions higher for both periods of comparison? How do the result change if the results from the PRMS models are not considered?

The analysis in section 2.3 provided reasons to retain the PRMS results.

3 The points raised in the table  are good recommendations for clarifying and expanding the text and we addressed them all in our revisions.

River Management Joint Operating Committee (RMJOC). (2017). NRNI Flows 1929-2008 Corrected 04-2017. Bonneville Power Administration. Retrieved from https://www.bpa.gov/p/Power-Products/Historical-Streamflow-Data/Pages/No-Regulation-No-Irrigation-Data.aspx

[revised manuscript text omitted]

---

## Author Response (AR2)

We are grateful to the reviewer and the editor for further pressing us to improve the paper. We summarize below the main issues, and provide our responses in blue. We also used the opportunity to to make other improvements including adding some more references and making some of the terminology clearer and more consistent.

Following are the three major comments/points/questions that should be addressed in the revisions.

1. How does the daily rainfall and resultant flood statistics compare with historic data? How well does the GCM downscaling match up and how well does the 4 hydrologic model generated flood event compare with the historic data for magnitude and timing. As mentioned in the paper, the previous work was based on annual and monthly flood statistics and the change in modelling timesteps require some sort of validation. Acknowledging that this basin has a lot of flow control, even the comparison of the upper reaches of the river system might be sufficient to gain some level of comparison on how well the four hydrologic model simulations compare with historic data.

We have carried out an extensive evaluation of our combined climate-hydrologic model system by comparing the GEV fits for the 40-member ensembles with the NRNI derived dataset (which accounts for reservoir operations, diversions, and evaporation) at a number of gauges on the Columbia, Snake, and Willamette. The previous version of the paper had a narrative summary of those comparisons but, in response to this comment, we have extended the analysis and included two new figures illustrating the performance of the modeling system for 10- and 100-year return periods to match the focus of the paper. New text describes the findings. Rather than evaluating the performance of the simulations of daily rainfall (or other factors influencing the hydrologic simulations) we simply focused on the outputs themselves.

2. As addressed in the conclusion of this paper, the question on how much of a contribution does the PRMS model results have on the increase in flood ratios needs to be addressed in some way as this paper suggests an increase in flood risk which is different to many other studies. A possible way that can be considered might be, were PRMS flood predictions higher for both periods of comparison? How do the result change if the results from the PRMS models are not considered?

Our more detailed examination of the modeling system's performance indicated that although PRMS is an outlier in some of the results (future change), it performs equally well in the evaluation (simulated past). Thus we can't a priori exclude it. We added some text in the discussion section.

Editor adds:

Verification of the current analysis as mentioned in section 2.3 need to be presented in the paper to substantiate the results. After re-reviewing the paper, results in Figure 7 suggests that further thought and attention should be give to the overall uncertainties of the results compared to the change in the flood index.

referee emphasises the need for being more explicit in your comparisons, and helping the reader understand the uncertainties in the results
We added some text in the discussion section.

[revised manuscript text omitted]

Body A

| Page 23: [2] Formatted | Mote, Philip W | 8/14/20 4:03:00 PM |
|---|---|---|

None, English (US)

| Page 23: [3] Formatted | Mote, Philip W | 8/14/20 4:03:00 PM |
|---|---|---|

None, English (US)

| Page 23: [4] Formatted | Mote, Philip W | 8/14/20 4:03:00 PM |
|---|---|---|

None, Font: Bold, English (US)

| Page 23: [5] Formatted | Mote, Philip W | 8/14/20 4:03:00 PM |
|---|---|---|

Body A, Line spacing:  single

| Page 23: [6] Formatted | Mote, Philip W | 8/14/20 4:03:00 PM |
|---|---|---|

None, Font: Times New Roman, Bold

| Page 23: [7] Formatted | Mote, Philip W | 8/14/20 4:03:00 PM |
|---|---|---|

Justified, Space After:  10 pt

| Page 23: [8] Formatted | Mote, Philip W | 8/14/20 4:03:00 PM |
|---|---|---|

None, Font: Times New Roman

| Page 23: [9] Formatted | Mote, Philip W | 8/14/20 4:03:00 PM |
|---|---|---|

None, Font: Times New Roman

| Page 23: [10] Deleted | Mote, Philip W | 8/14/20 4:03:00 PM |
|---|---|---|

| Page 23: [11] Formatted | Mote, Philip W | 8/14/20 4:03:00 PM |
|---|---|---|

None, Font: Times New Roman

| Page 23: [12] Formatted | Mote, Philip W | 8/14/20 4:03:00 PM |
|---|---|---|

None, Text Outline

| Page 23: [13] Formatted | Mote, Philip W | 8/14/20 4:03:00 PM |
|---|---|---|

None, Text Outline

| Page 23: [14] Formatted | Mote, Philip W | 8/14/20 4:03:00 PM |
|---|---|---|

None, English (US), Text Outline

| Page 23: [15] Formatted | Mote, Philip W | 8/14/20 4:03:00 PM |
|---|---|---|

Correspondence, Line spacing:  1.5 lines

| Page 23: [16] Formatted | Mote, Philip W | 8/14/20 4:03:00 PM |
|---|---|---|

None, German, Text Outline

| Page 23: [17] Formatted | Mote, Philip W | 8/14/20 4:03:00 PM |
|---|---|---|

Body C, Justified, Line spacing:  1.5 lines

| Page 23: [18] Formatted | Mote, Philip W | 8/14/20 4:03:00 PM |
|---|---|---|

| Page 23: [19] Formatted | Mote, Philip W | 8/14/20 4:03:00 PM |
|---|---|---|

None, Font: 10 pt, English (US)

| Page 23: [19] Formatted | Mote, Philip W | 8/14/20 4:03:00 PM |
|---|---|---|

None, Font: 10 pt, English (US)

| Page 23: [19] Formatted | Mote, Philip W | 8/14/20 4:03:00 PM |
|---|---|---|

None, Font: 10 pt, English (US)

| Page 23: [19] Formatted | Mote, Philip W | 8/14/20 4:03:00 PM |
|---|---|---|

None, Font: 10 pt, English (US)

| Page 23: [19] Formatted | Mote, Philip W | 8/14/20 4:03:00 PM |
|---|---|---|

None, Font: 10 pt, English (US)

| Page 23: [20] Formatted | Mote, Philip W | 8/14/20 4:03:00 PM |
|---|---|---|

None, Font: 10 pt

| Page 23: [21] Formatted | Mote, Philip W | 8/14/20 4:03:00 PM |
|---|---|---|

None, German, Text Outline

| Page 23: [22] Formatted | Mote, Philip W | 8/14/20 4:03:00 PM |
|---|---|---|

Correspondence, Line spacing:  1.5 lines

| Page 23: [23] Formatted | Mote, Philip W | 8/14/20 4:03:00 PM |
|---|---|---|

Body C, Justified, Line spacing:  1.5 lines

| Page 23: [24] Formatted | Mote, Philip W | 8/14/20 4:03:00 PM |
|---|---|---|

None, Font: 10 pt

| Page 23: [25] Formatted | Mote, Philip W | 8/14/20 4:03:00 PM |
|---|---|---|

None, Font: 10 pt

| Page 23: [25] Formatted | Mote, Philip W | 8/14/20 4:03:00 PM |
|---|---|---|

None, Font: 10 pt

| Page 23: [26] Formatted | Mote, Philip W | 8/14/20 4:03:00 PM |
|---|---|---|

None, Font: 10 pt, English (US)

| Page 23: [26] Formatted | Mote, Philip W | 8/14/20 4:03:00 PM |
|---|---|---|

None, Font: 10 pt, English (US)

| Page 23: [26] Formatted | Mote, Philip W | 8/14/20 4:03:00 PM |
|---|---|---|

None, Font: 10 pt, English (US)

| Page 24: [27] Formatted | Mote, Philip W | 8/14/20 4:03:00 PM |
|---|---|---|

None, Font: 10 pt

| Page 24: [28] Formatted | Mote, Philip W | 8/14/20 4:03:00 PM |
|---|---|---|

Correspondence, Line spacing:  1.5 lines

| Page 24: [29] Formatted | Mote, Philip W | 8/14/20 4:03:00 PM |
|---|---|---|

None, German, Text Outline

| Page 24: [31] Formatted | Mote, Philip W | 8/14/20 4:03:00 PM |
|---|---|---|

Body C, Justified, Line spacing: 1.5 lines

| Page 24: [32] Formatted | Mote, Philip W | 8/14/20 4:03:00 PM |
|---|---|---|

None, Font: 10 pt

| Page 24: [32] Formatted | Mote, Philip W | 8/14/20 4:03:00 PM |
|---|---|---|

None, Font: 10 pt

| Page 24: [33] Formatted | Mote, Philip W | 8/14/20 4:03:00 PM |
|---|---|---|

None, Font: 10 pt

| Page 24: [34] Formatted | Mote, Philip W | 8/14/20 4:03:00 PM |
|---|---|---|

None, Font: 10 pt, English (US)

| Page 24: [35] Formatted | Mote, Philip W | 8/14/20 4:03:00 PM |
|---|---|---|

Body C, Line spacing: 1.5 lines

| Page 24: [36] Formatted | Mote, Philip W | 8/14/20 4:03:00 PM |
|---|---|---|

None, Font: 10 pt, English (US)

| Page 24: [37] Formatted | Mote, Philip W | 8/14/20 4:03:00 PM |
|---|---|---|

None, Font: (Default) Calibri, 10 pt, English (US)

| Page 24: [38] Deleted | Mote, Philip W | 8/14/20 4:03:00 PM |
|---|---|---|

| Page 24: [39] Formatted | Mote, Philip W | 8/14/20 4:03:00 PM |
|---|---|---|

None, Font: (Default) Calibri, 10 pt

| Page 24: [40] Formatted | Mote, Philip W | 8/14/20 4:03:00 PM |
|---|---|---|

None, Font: 10 pt, English (US)

| Page 24: [41] Formatted | Mote, Philip W | 8/14/20 4:03:00 PM |
|---|---|---|

Body C, Justified, Line spacing: 1.5 lines

| Page 24: [42] Formatted | Mote, Philip W | 8/14/20 4:03:00 PM |
|---|---|---|

None, Font: 10 pt, English (US)

| Page 24: [43] Formatted | Mote, Philip W | 8/14/20 4:03:00 PM |
|---|---|---|

None, Font: 10 pt, English (US)

| Page 24: [44] Formatted | Mote, Philip W | 8/14/20 4:03:00 PM |
|---|---|---|

None, Font: 10 pt, English (US)

| Page 24: [45] Formatted | Mote, Philip W | 8/14/20 4:03:00 PM |
|---|---|---|

None, Text Outline

| Page 24: [46] Formatted | Mote, Philip W | 8/14/20 4:03:00 PM |
|---|---|---|

None, Text Outline

| Page 24: [47] Formatted | Mote, Philip W | 8/14/20 4:03:00 PM |
|---|---|---|

None, Text Outline

---

## Author Response (AR3)

Dear Dr. Viviroli,

Thank you for your continued attention to the review process for our paper. We believe we have addressed the concerns expressed by Reviewer 4

"*consider the change in flood index within the total accumulated and cascading uncertainty of the results to see if the claim of a contradicting result is still valid.*"

As well as your guidance

and even though you do make mention of limitations, there is strong need for clarification. In this sense, discussing potential weaknesses and their impact on your findings clearly would be a considerable added value to the paper. However, it might not be possible to accept the manuscript for publication in HESS unless that is done comprehensively and throughout the manuscript (including abstract)

We have added comments on the limitations of our study in the abstract, introduction, and in sections 2.1, 2.3, 3.1, and 4.

or, alternatively, if you can provide a clear rebuttal of the concerns, also to be reflected in the manuscript appropriately.

While the errors in future projections of climate change are always unknowable in advance, we contend that we have exceeded the standards set by most if not all other papers of this type in quantifying the sources of uncertainty. The flood validation figures added in response to the reviewer already set this paper apart from its predecessors, and moreover several figures in our paper (viz., Figs 7-10) explicitly show the effects of these choices in the final results which other studies do not do.

We contend, here and in the revisions, that the use of a wide variety of climate scenarios, of two different downscaling approaches, of four different hydrologic model configurations, and finally of 10- and 100-year return periods (as well as other methodological choices taken but not included, as being shown by our initial analysis to be largely immaterial to the results), should produce *more,* not less, heterogeneity of results compared with previous studies which all lacked this degree of complexity and provided little if any scrutiny of past performance of the modeling systems used there.

In fact, we struggle to see how one could design and carry out an estimate of future flooding that would improve substantially on this one, without using at least ten times the computing and personnel resources as we (or any typical group of authors) have at our disposal.

[revised manuscript text omitted]

---

## Author Response (AR4)

We have made all the corrections the editor specified, namely three sentence edits and the removal of references that are not referenced in the text. We also made the suggested corrections to Figure 1.

**Formatted**

[revised manuscript text omitted]